# Magnetic Silica-Coated Iron Oxide Nanochains as Photothermal Agents, Disrupting the Extracellular Matrix, and Eradicating Cancer Cells

**DOI:** 10.3390/cancers11122040

**Published:** 2019-12-17

**Authors:** Jelena Kolosnjaj-Tabi, Slavko Kralj, Elena Griseti, Sebastjan Nemec, Claire Wilhelm, Anouchka Plan Sangnier, Elisabeth Bellard, Isabelle Fourquaux, Muriel Golzio, Marie-Pierre Rols

**Affiliations:** 1Institute of Pharmacology and Structural Biology, 205 Route de Narbonne, 31400 Toulouse, France; elena.griseti@ipbs.fr (E.G.); Elisabeth.Bellard@ipbs.fr (E.B.); muriel.golzio@ipbs.fr (M.G.); Marie-Pierre.Rols@ipbs.fr (M.-P.R.); 2Department for Materials Synthesis, Jožef Stefan Institute, Jamova cesta 39, 1000 Ljubljana, Slovenia; sebastjan.nemec@ijs.si; 3Faculty of Pharmacy, University of Ljubljana, Askerceva cesta 7, 1000 Ljubljana, Slovenia; anouchka.plan@gmail.com; 4Laboratoire Matière et Systèmes Complexes (MSC), UMR 7057, Bâtiment Condorcet, Université Paris Diderot, 10 rue Alice Domon et Léonie Duquet, 75205 Paris, France; claire.wilhelm@univ-paris-diderot.fr; 5Centre de Microscopie Electronique Appliquée à la Biologie (CMEAB), Faculté de Médecine Rangueil, 133 Route de Narbonne, 31400 Toulouse, France; isabelle.fourquaux@univ-tlse3.fr

**Keywords:** magnetic silica-coated iron oxide nanochains, nanoparticles, photothermal treatment, hyperthermia, cancer, collagen, cellular microenvironment

## Abstract

Cancerous cells and the tumor microenvironment are among key elements involved in cancer development, progression, and resistance to treatment. In order to tackle the cells and the extracellular matrix, we herein propose the use of a class of silica-coated iron oxide nanochains, which have superior magnetic responsiveness and can act as efficient photothermal agents. When internalized by different cancer cell lines and normal (non-cancerous) cells, the nanochains are not toxic, as assessed on 2D and 3D cell culture models. Yet, upon irradiation with near infrared light, the nanochains become efficient cytotoxic photothermal agents. Besides, not only do they generate hyperthermia, which effectively eradicates tumor cells in vitro, but they also locally melt the collagen matrix, as we evidence in real-time, using engineered cell sheets with self-secreted extracellular matrix. By simultaneously acting as physical (magnetic and photothermal) effectors and chemical delivery systems, the nanochain-based platforms offer original multimodal possibilities for prospective cancer treatment, affecting both the cells and the extracellular matrix.

## 1. Introduction

Iron oxide nanoparticles (IONPs) are among the most popular and most extensively studied inorganic nanoparticles, which have enabled a series of distinct therapeutic approaches in various biomedical domains [1].

Iron oxide nanoparticles were historically used as contrast agents for magnetic resonance imaging (MRI), mostly to detect liver metastases [2]. Subsequently, IONPs use evolved with the development of MRI-based technologies, enabling the in vivo tracking of single cells [3]. The magnetic responsiveness of IONPs [4] paved the way to magnetic cell manipulation approaches.

In order to be useful in biomedicine, nanoparticles should be superparamagnetic. Superparamagnetism is a form of magnetism where magnetization randomly flips at room temperature. In the absence of an external magnetic field, the overall magnetization of a group of superparamagnetic nanoparticles (smaller than 20 nm for iron oxides) is zero, because the magnetic moment of the nanoparticles is randomly distributed. The weakness of the attractive magnetic interactions among the superparamagnetic nanoparticles does not to allow them to magnetically aggregate. This magnetic behavior is crucial for the preparation of ferrofluids, which can therefore be applied to biological systems, because superparamagnetic nanoparticles remain well dispersed. Nevertheless, when an external magnet is applied, all magnetic moments align in the same direction, leading to a net magnetization. Nevertheless, individual superparamagnetic nanoparticles are too small for their effective translational movement even in the strongest magnetic fields. Therefore, in view of magnetic manipulation, iron oxides can be loaded in liposomes, confined within cell-derived vesicles or loaded within cells. When IONPs are loaded within the cells, the latter become magneto-responsive, and can thus migrate along a magnetic field gradient, allowing distal guidance of nanoparticle-loaded cells to the site of therapeutic interest [5].

Alternatively, to preserve superparamagnetism and achieve strong magnetic responsiveness, we here present a method by which we group a number of small superparamagnetic nanoparticles into larger and defined nanoparticle clusters. Such clusters are highly magnetically responsive and therefore form chain-like structures once exposed to magnetic field. In order to obtain permanent chain-like structures—the magnetic “nanochains”—we optimized the synthesis protocol. The nanochains are thus obtained when a suspension of superparamagnetic nanoparticle clusters is exposed to a defined magnetic field, while silica is added to fixate the clusters into permanent nanochains. Such nanochains are superparamagnetic, with superior magnetic responsiveness due to much larger magnetic moment (larger magnetic volume) than that of individual nanoparticle clusters.

In addition to magnetic targeting, other biomedical applications are applicable to IONPs. When magnetic nanoparticles are submitted to a high frequency alternating magnetic field (AMF) with suitable amplitude, the magnetic energy is transduced to nanoparticle heating, which dissipates into the surroundings. This phenomenon is exploited in magnetic hyperthermia [6], an experimental cancer treatment, where magnetic nanoparticles are locally injected and heated, in order to destroy unresectable tumors. Magnetic hyperthermia has an important advantage over other, more common hyperthermia treatments, such as the ones generated by microwaves, radiofrequency, or ultrasound. This advantage relies on the potential to heat solid tumors from the inside, preventing the heating of bystander tissues, located between the energy source and the target zone. Nevertheless, the main drawbacks for magnetic hyperthermia are: (1) the requirement of injecting high amounts of IONPs in order to obtain significant heating, thus requiring local (intra-tumoral) nanoparticles injections, and (2) the necessity to keep nanoparticles outside the cells, as intracellular processing dramatically decreases nanoparticle heating potential [7].

For a long time, magnetic hyperthermia was considered as the only nanoparticles-mediated thermal modality for cancer treatment. However, a few years ago, their remarkable conversion of light to heat in the near infrared window was demonstrated [8]. Photothermal therapy is a treatment that relies on the conversion of the energy of light into thermal energy. This phenomenon occurs because iron oxides are semiconductors, and have a small band gap between valence and conduction electrons. This band gap can be bridged by excitation, provided by the optical energy (the photons) of a laser beam. After excitation, valence electrons stay in the conduction band for a limited time, after which they fall back to the valence position. When this fall occurs, heat is emitted [9]. As light sources, such as lasers, can be accurately focused, the photothermal treatment can be limited only to the zone of therapeutic interest. The main advantage of photothermia over magnetic hyperthermia is that the treatment could be efficient even after cellular internalization of nanoparticles and at lower local nanoparticles concentrations [7].

Photothermia has been used in preclinical research, and mainly relies on heat generation mediated by gold [10], silver [11], and copper [12] nanoparticles or carbon nanotubes [13] and graphene [14]. These nanomaterials have a different mechanism for heat generation than IONPs. In metals, such as gold, silver, and copper, the light to heat conversion occurs when light interacts with conduction electrons on the surface of metallic nanoparticles [15]. In carbon-based nanoparticles, the delocalized electrons absorb light and the energy is converted to vibrations of the C-C reticule, which is released as heat when the vibrational states decay [15].

Photothermal therapy using gold nanoparticles already reached clinical trials (ClinicalTrials.gov Identifiers: NCT01270139 and NCT01436123), yet, as these materials might be extremely bio-persistent [16] and can potentially be toxic [17,18], we suggest alternative bio-compatible materials. Among them, iron oxides appear particularly attractive. The median lethal dose of intravenously applied citrate-coated IONPS (diameter 8.6 nm) to mice is very high, and was reportedly greater than 949 mg (17 mmol) Fe/kg body weight [19]. Indeed, once administered by the intravenous route, IONPs mainly accumulate in the liver. There they may induce oxidative stress, which results in an increased level of liver enzymes. Nevertheless, these changes are transient, and they do not lead to significant adverse reactions [20]. When synthetic nanoparticles degrade, the iron loads into ferritin proteins and integrates iron’s physiological pathways of iron re-use or elimination [21]. Alternatively, the iron, released from synthetic nanoparticles, can be re-assembled in cells to form nanoparticles anew, and prevent the toxicity of ferrous iron [22].

While IONPs potential for MRI, magnetic targeting, and magnetic hyperthermia have been broadly documented in the literature, IONP-induced photothermia recently emerged [8,23].

With the aim to create bio-compatible, magnetically guidable, and photothermally responsive nanoparticles with long-term multimodal imaging properties, we intentionally prepared and optimized the functionalization protocol, as well as biologically tested magnetic silica-coated iron oxide nanochains [24]. The chains have an iron-oxide core and a silica shell, within which we covalently linked a fluorescent dye, allowing fluorescence imaging follow-up. The core conveys magnetic and photothermal properties, and the shell provides both (i) a large platform for nanoparticles surface functionalization and (ii) a porous compartment, which can covalently bind molecular components, such as dyes or drugs. The nanochains, which are presented in this study represent a promising and unique nanomaterial, possessing superparamagnetism, superior magnetic responsiveness, and good photothermal properties, which have been rarely combined in single nanostructures.

Cancerous cells and their microenvironment both play a pivotal role in cancer development, progression, and resistance to treatment [25]. In this regard, the nanochains, presented herein, which heat and thus simultaneously affect cellular and environmental components, could radically improve the therapeutic outcome. Moreover, one of the very important advantages of the magnetic nanochains is their superior magnetic responsiveness. The latter could be used for all applications where magnetic guidance is desired, such as magnetic drug delivery or magnetic targeting.

## 2. Results

### 2.1. Nanochains Characterization

Short, less than 1 µm long, anisotropic by shape, and superparamagnetic nanochains were synthesized by magnetic assembly of few (5 ± 1.4) nanoparticle clusters (Figure 1A, Appendix A). The size of nanoparticle clusters coated with a 3-nm-thick silica shell was measured from TEM micrographs (>100 nanochains counted) and measured 90 ± 28 nm. The synthesis of nanoparticle clusters is based on self-assembly of 70 ± 14 superparamagnetic iron oxide (maghemite) nanoparticles within the size range of 10.4 ± 1.4 nm [24,26,27]. The core–shell nature of the nanochains, with the closely packed maghemite nanoparticles cluster cores and the amorphous silica shell, can be clearly distinguished in TEM micrographs (Figure 1A). Functionalized fluorescent nanochains showed superparamagnetic properties, excellent colloidal stability, and high magnetic responsiveness (Appendix A). When the suspension was placed on a magnet, the nanochains (Figure 1B left) readily assembled into slightly larger anisotropic nanochain bundles that could be detected by optical microscopy (Figure 1B right). These bundles are disaggregated as soon as the external magnet is removed. The nanochains’ saturation magnetization Ms is ~37 Am^2^kg^−1^ (Figure 1C). The efficiency of the functionalization was assessed using the Kaiser test, FTIR-ATR (Appendix A), and indirect methods such as the zeta-potential measurements of the fluorescent silica-coated nanochains (RB-nanochains), fluorescent amino-functionalized nanochains (RB-nanochains-NH_2_), and fluorescent carboxyl functionalized nanochains (RB-nanochains-COOH) in aqueous suspensions (Figure 1D). The accessible primary amines on the surface of the RB-nanochains-NH_2_ was quantified by Kaiser test. The spectroscopically determined mean value is 145 µmol per gram of the RB-nanochains-NH_2_. A FTIR-ATR surface analysis method confirmed the transfer of surface amines of the RB-nanochains-NH_2_ into carboxyl groups on the surface of the RB-nanochains-COOH (see Appendix A). The spectrum of the RB-nanochains-COOH shows the distinctive bands at wavenumbers 1703.1 cm^−1^, 1442.9 cm^−1^, and 1402.7 cm^−1^, confirming the presence of carboxyl group and amide bond. However, the primary amines of the RB-nanochains-NH_2_ are not clearly visible because they are overlapped with intensive silanol OH of silica at wavenumbers above 3000 cm^−1^. The electrophoretic mobilities of the RB-nanochains, RB-nanochains-NH_2_, and RB-nanochains-COOH were measured as the function of the operational pH. The silica surface shows a relatively acidic character, because its structure comprises negatively charged –OH groups at pH values above the isoelectric point (IEP) at ~ pH 3. The zeta-potential curve of pristine silica-coated nanochains reaches negative values of less than 15 mV at pH above 7. After amine functionalization, the zeta-potential of the RB-nanochains-NH_2_ changed significantly and reached the value of ~ 6 mV at physiological pH 7.4. More pronouncedly, after carboxyl functionalization, the zeta potential significantly decreases to negative values, reaching negative values of less than ~ 40 mV at pH above 7. The high absolute values of the zeta-potential provide strong electrostatic repulsive forces between the nanochains, and result in a good colloidal stability of the suspension in neutral and alkaline conditions [28]. Therefore, the carboxyl-functionalized nanochains (nanochains-COOH and RB-nanochains-COOH) were chosen for further investigation. The heating efficiency was evaluated in nanochains suspensions up to [Fe] = 130 mM (7.3 g_Fe_/L). The magnetic hyperthermia (MHT) yield after exposure to an alternating magnetic field was low even at the highest concentrations (130 mM): a 400 s exposure of the nanochains suspension led to a temperature increase of only 3.5 °C ± 0.2 °C. Heating potential can be evaluated using the mass-normalized specific absorption rate (SAR) parameter, expressed in W per gram of iron. Herein, the MHT SAR reached 25 ± 9 W/g_Fe_. Conversely, the RB-nanochains-COOH suspension provided a high photothermal yield (Figure 1E) upon laser excitation at 808 nm and laser power density of 0.3 W/cm^2^. A 400 s exposure of the same suspension led to a temperature increase of about 15.7 ± 2 °C (Figure 1E), and corresponded to a SAR of 202 ± 25 W/g_Fe_. Indeed, the thermal yield is proportional to the laser power density, and reached a temperature increase of 30 ± 2 °C for the same suspension volume (100 µL) and iron concentration (130 mM), irradiated at λ = 808 nm at a laser power density of 1 W/cm^2^.

### 2.2. Cell Loading

The cells were efficiently loaded with RB-nanochains-COOH, as shown in Figure 2A–D. RB-nanochains-COOH were internalized to the highest degree by the largest cells—the fibroblasts, while HCT-116 (wild type), HCT-116-GFP, and HeLa GFP-Rab7 internalized comparable quantities of chains, as assessed by average fluorescence intensity measurements. The mean red fluorescence intensity of internalized chains was 105 ± 6 for fibroblasts, 72 ± 1 for HeLa GFP-Rab7 cells, 74 ± 2 for HCT-116-GFP cells, and 71 ± 2 for HCT-116 wild type cells, respectively. Within the cells, the nanochains localized within endosomes, as evidenced in Figure 2E,F, showing HeLa cells expressing a green fluorescent rab7 protein, which is associated to early and late endosomes [25]. In HCT-116 wild type cells, which internalized the smallest amount of RB-nanochains-COOH, we quantified the iron load by single cell magnetophoresis, and determined an uptake of 17.3 ± 2.6 pg of iron per cell (Figure 2G). The mass of a single nanochain composed of 5 nanoparticle clusters is estimated to be 2.1 × 10^−14^ g. Since iron represents approximately 45% of nanochains, this equals 0.9 × 10^−14^ g of iron per chain. We thus estimate around 1800 ± 270 nanochains per cell in cancer cells. The uptake comparison among different cell types, used in this study, is shown in Figure 2H. While cancer cells internalized comparable amounts of RB-nanochains-COOH, the fibroblasts internalized about 30% more nanochains.

### 2.3. Short and Long-Term Toxicity Assessment

Cell incubation with RB-nanochains-COOH (performed at 5 mM extracellular concentration of iron) did not alter cell viability of HCT116, HCT116-GFP, HeLa GFP-Rab7, and normal human dermal fibroblasts, as assessed with the Trypan Blue Exclusion Test, performed 24 h after cell loading. The viability of control and loaded cells was above 90%, respectively. In order to assess the long-term effect of RB-nanochains-COOH on cell survival and proliferation, we performed two distinct tests. The first approach involved a clonogenic test (Figure 3A,B). In the case of HCT-116, HCT-116-GFP, and HeLa-Rab7-GFP cells, the cells exhibited a colony-forming capacity of 83%, 90%, or 112%, respectively. In the case of normal dermal fibroblasts, the colony-forming capacity of loaded cells was of about 70% compared to control cells.

The second approach involved the formation of cellular spheroids with unloaded (control) or loaded cells and the monitoring of spheroid growth over 9 days (Figure 4A). Unloaded- and loaded-cells’ spheroids exhibited comparable growth (Figure 4B).

### 2.4. Nanochain Distribution within Tumor Cells and Dermal Fibroblasts Spheroids

The distribution of RB-nanochains-COOH within the spheroids was assessed by a TEM investigation (Figure 5). In cellular spheroids composed of normal dermal fibroblasts, the nanochains are localized in endo-lysosomal compartments within the cells, and within the extracellular matrix, while in MCS made of cancer cells the nanochains are exclusively found in the endo-lysosomal compartments within cells.

Within the extracellular matrix of spheroids, made of normal dermal fibroblasts, the RB-nanochains-COOH were found confined in the extracellular vesicles (Figure 6A) and non-confined, surrounded by fibrillary structures of the extracellular matrix (Figure 6B).

### 2.5. Photothermia

After ascertaining that nanochains are well tolerated and not toxic to different cell types in vitro, we evaluated their potential for photothermal therapy. Plated cells, loaded with nanochains-COOH, or engineered cell sheets loaded with RB-nanochains-COOH, were exposed to the laser source operating within the multiphoton microscope. When loaded cells were exposed to the laser, all of them almost instantaneously underwent cell death. Figure 7A shows a characteristic example of loaded cells, which after nanochain-COOH heating underwent cell death, as evidenced by propidium iodide uptake (Figure 7B). Cell sheets, made of normal dermal fibroblasts, have a rich collagenous matrix. In the absence of RB-nanochains-COOH, the cells and their matrix were not altered after laser exposure under our experimental conditions (Figure 7C). When RB-nanochains-COOH were added to cell sheets (Figure 7D), they distributed transversally within the sheets and were found within both the cells and the collagen matrix (Figure 7E). While at low laser power (20 mW) the collagen remained unaltered, when the laser power was increased to 33 mW, the nanochains heated and melted the adjacent collagen fibers. The heating was well localized and was limited only to zones rich in nanochains and exposed to the higher laser power, as evidenced in Figure 7F. Zones where nanochains were not present and control cell sheets were not altered after 33 mW laser exposure.

## 3. Discussion

In this work we characterized a novel platform made of superparamagnetic iron oxide nanoparticle assemblies—silica-coated iron oxide nanochains [24]—that could have therapeutic potential once photoactivated.

Nanochains have high magnetic responsiveness, as we previously reported [24] and evidenced in Appendix A. Herein, we labeled the nanochains with rhodamine, which was covalently bound into the silica shell. This fluorescent labelling allows nanochains tracking by fluorescence imaging, while the surface of nanochains was functionalized with carboxyl groups (Figure 1), which are advantageous for cell-uptake, and allow endowing the cells with a high amount of nanoparticles (Figure 2). Presented nanostructures were characterized by different physico-chemical tests. Dynamic light scattering is frequently applied for determination of colloid hydrodynamic size, yet this analysis is designed for isotropic (spherical) particles. This method is therefore useless for characterization of nanochains due to their highly anisotropic shape. Alternatively, the behavior of colloidal stability in suspension could be estimated by monitoring the spontaneous sedimentation of the colloids over time. The RB-nanochains-COOH remained in suspension for at least three weeks (Appendix A) and then only gradual could color change be seen, indicating a good stability for such relatively large “colloids”. The RB-nanochains-COOH sediment completely due to gravity in two months, but they can be easily redispersed with a single gentle shake by hand. Another indirect method to determine the colloidal stability is the measurement of the RB-nanochains-COOH zeta potential, where high absolute values (over ± 30mV) at physiologically relevant pH 7.4 suggest good colloidal stability. In contrast, RB-nanochains are slightly negatively charged (−12 ± 3 mV) and RB-nanochains-NH_2_ are slightly positively charged (6 ± 5 mV) at pH 7.4. These values are both inappropriate for achieving a suitable colloidal stability, but useful in biological settings. We therefore focused on carboxyl-functionalized nanochains (nanochains-COOH and RB-nanochains-COOH) for our in vitro studies.

Cell-loaded nanoparticles equaled a minimum of about 15 pg of iron per cell, as measured in HCT-116 cells (Figure 2), which were the smallest cells among the ones we used in this study and exhibited the lowest average red-fluorescence intensity. The quantity of iron per cell was determined by magnetophoresis. In the same way as for spherical nanoparticles, once internalized by cells, nanochains end up in large endo-lysosomal compartments [29], showing no shape anisotropy. Therefore, in the same way as for other nanoparticle-loaded cells, the translational movement of magnetic-nanochain-loaded cells derives from the overall load, which provides a global magnetic moment of the cell. Indeed, as the proportion of internalized material depends on cell size [29], fibroblasts internalized higher amounts of nanoparticles (Figure 4) and fluoresced about 30% more than cancerous cells.

In this study, the in vitro toxicity of nanochains was evaluated for the first time. The nanochains were not toxic to cancerous and non-cancerous cells (normal dermal fibroblasts, isolated from a skin biopsy), as assessed by three distinct approaches (trypan blue exclusion test, colony-forming ability, and RB-nanochain-COOH-loaded cells spheroid growth). While iron oxides are generally considered as safe and biocompatible, the toxicity assessment was indeed necessary because of the size and anisotropy of tested nanomaterials. In this regard, the toxicity was evaluated at different time points after cellular uptake of RB-nanochains-COOH. At first, we thus performed the toxicity assessment 24 h after cell loading. Cell viability (and thus short-term toxicity) is commonly evaluated using metabolic assays, which rely on read-out principles that include absorbance (colorimetry), fluorescence, or luminescence. Nevertheless, as iron oxides absorb light, colorimetric assays (such as MTT or MTS) or bio-luminescent tests (such as the ATP-based assays) might lead to false readouts, as studies reported [30]. Moreover, iron oxides can act as fluorescence quenchers [31,32,33], therefore redox-based viability tests, using fluorescent dyes, were also considered inappropriate for our study. We therefore assessed the viability with the trypan blue exclusion test [34]. This test relies on the capacity of viable cells to exclude the dye and the viability was assessed by direct visualization of cells and manual counting. As trypan blue exclusion only provides information about early events that occur after nanoparticles uptake, we used 2 additional tests, by which we evaluated the toxicity in the long term (7 days and beyond). Our results indicate that nanochains are well tolerated by the cells. In the case of fibroblasts, we could observe a 30% decrease in colony-forming ability. This is probably due to two factors. Firstly, the fibroblasts have a surface that is relatively larger than the ones of cancerous cells, therefore they internalize more nanoparticles at the same extracellular nanoparticles concentration [35] (Figure 2H). Secondly, when cells divide, the nanoparticles load divides between daughter cells [36]. When cells have a short doubling time, which is the case of cancer cells, the iron load quickly splits and the burden of oxidative stress diminishes. The dermal fibroblasts which were used in this study had a doubling time of about 40 h, which was twice as much as the doubling time of cancer cells. Thus, fibroblasts were more exposed to the strain of oxidative stress, due to high, persistent, and prolonged iron presence. Moreover, studies show that gene expression of ferritin, the iron storage protein, and other proteins involved in iron homeostasis are upregulated after cell internalization of iron oxide nanoparticles [37], and this is even more emphasized in different cancers, as numerous studies suggest [38]. This explains why the colony-forming ability of cancer cells, used in this study, was less affected: cancer cells adapted easier than fibroblasts. In order to verify if nanoparticles could have a pro-proliferative effect, an additional test was made, and consisted in making cellular spheroids with cells, which were loaded with RB-nanochains-COOH. Spheroid growth was subsequently followed over 9 days (Figure 4). A modest but significant difference was only noted in the case of dermal fibroblasts, where spheroids made with particle-loaded cells were slightly larger than the spheroids made with unloaded cells. We assume that this difference is due to the overall volume of nanoparticles, internalized by cells, rather than to any potential cell growth stimulation, as the slopes of loaded and unloaded MCS growth curves are equivalent. As the volumes of cancer cell spheroids were much larger, the slight volume increase, which might be due to nanoparticles presence, affected the final volume to a much smaller extent, and the differences between loaded and non-loaded cells spheroids sizes were not significant. This result also allowed us to exclude a pro-proliferative effect induced by nanochains.

The distribution of nanochains was assessed on the ultrastructural level with transmission electron microscopy (TEM), by which we analyzed cells in 2D (at D1 post nanochains uptake—Figure 2) and 3D cultures (cellular spheroids—Figure 5, Figure 6 and Appendix A), as well as cell sheets (Figure 7). Nanochains are localized within cancer cells in multicellular spheroids, while in spheroids made of fibroblasts and in cell sheets the nanochains are distributed within cells and the extracellular matrix. On the ultrastructural level, the cells exhibited a normal morphology and we did not observe any signs of autophagy [28]. Nanochain load was much more prominent in spheroids made of fibroblasts, as these cells internalize more nanoparticles and have a smaller proliferation rate (as they exhibit contact inhibition when grown in 3D structures). This also explains why cell spheroids made of fibroblasts remain much smaller than the spheroids made of cancer cells and explains the auto-fluorescence of control fibroblasts spheroids, which started fluorescing (at λ_Ex_ = 560/40 nm and λ_Em_ = 630/75 nm) from day 5 onwards (Figure 4E first column). Dead cells have an increased autofluorescence, which is due to a decreased metabolic activity, and increased levels of denatured proteins [39,40]. Moreover, while cancerous cells divided their nanochain load and kept it within the cells, fibroblasts “expelled” nanochains into the matrix, (Figure 6), where chains were either contained within extracellular vesicles [41] or were not enclosed in membrane-derived structures. Based on this evidence, we stipulate that normal cells, which internalize more nanoparticles, have a smaller proliferation rate and a higher death rate (with respect to the overall number of cells constituting the spheroids), and generate more extracellular vesicles, which delocalize nanoparticles outside the cells into the extracellular matrix.

On the nanoparticle level, the nanochains did not undergo any major structural disintegration (Figure 6 and Appendix A). Their architecture at day 11 (when the spheroids were fixed and processed for TEM, as evidenced in multicellular spheroids in Figure 5, Figure 6 and Appendix A), remained comparable to their initial structure, which was observed in freshly loaded cells (Figure 2B) when the cells were seeded to form spheroids. Indeed, over time the chains are expected to degrade, as silica metabolizes to silicic acid [42], which is excreted in the urine [43]. On the other hand, the iron, stemming from iron oxides, is integrated into the metabolic pathway of endogenous iron [44].

Finally, our study demonstrates that presented nanochains have an excellent photothermal yield (Figure 1E), which can lead to both cell death (Figure 7B) and the melting of the extracellular matrix (Figure 7F). Most anticancer agents target cancerous cells. Nevertheless, the cellular matrix, which constitutes the tumor microenvironment, provides cancer cells with nutriments and structural support, as well as represents a substantial barrier to anticancer agent penetration. Moreover, the mechanical compression exerted by the tumor matrix was recognized to induce the metastatic phenotype in tumor cells [45].

Heat-induced cell death was previously reported for magnetic hyperthermia [6]. Nevertheless, magnetic heating can only occur when magnetic nanoparticles have enough freedom to rotate. When magnetic nanoparticles are submitted to an alternating magnetic field, they orient their magnetic moments. Magnetic energy is thus dissipated through nanoparticle movement, either by nanoparticle rotation due to Brownian relaxation or by the rotation of the magnetic moments within nanoparticles’ cores due to Néel relaxation [7]. Therefore, if magnetic nanoparticles are confined, their magneto-thermal yield becomes much lower [7]. Nanoparticle confinement, for example, occurs within endo-lysosomes. In order to prevent such confinement after cellular internalization and in order to prevent nanoparticles dissemination throughout the body, the current therapeutic strategy (used by Magforce, the company that implemented the clinical use of magnetic hyperthermia) relies on the use of an aminosilane coating, which allows the formation of “stable deposits” within the tumoral tissue [46]. The nanochains, which are described in this study, are made of iron oxide nanoparticles confined within a silica shell. The low magnetic hyperthermia yield (ΔT = 3.5 °C) was thus not surprising, and magnetic hyperthermia was not considered as a modality of choice. In contrast, the nanochains appeared to be excellent candidates for photothermia [7], exhibiting a high photothermal yield, even at low and physiologically useful laser power density (ΔT = 30 °C at light wavelength of 808 nm and laser power density of 1 W/cm^2^ and ΔT = 15.7 °C at a laser power density of 0.3 W/cm^2^). The photothermal functionality is particularly attractive, because the thermal yield does not decrease after endo-lysosomal internalization [7]. Nanochains can thus heat the surrounding environment when they are located outside and inside the cells (Figure 7).

Iron oxide nanoclusters, made of Fe_3_O_4_ nanocrystals interconnected by amorphous matrix bridges, were previously reported as efficient photothermal agents [23]. While this study was among the first to evidence that nanoclusters generate more heat than individual, non-clustered nanoparticles, the photothermal effect was obtained using a laser power density of 5 W/cm^2^ [23]. Such laser power is high and detrimental to the tissue, and far exceeds the Maximum Permitted Exposure for the skin, as suggested by the American National Standards Institute Z136. In comparison, the nanochains could yield significant heat (more than 15 °C) at a much lower, and physiologically tolerable, laser power density of 0.3 W/cm^2^.

The heating of nanochains was sufficient to induce cell death and to melt the collagen matrix, as evidenced in Figure 7. We have previously reported on collagen fibers slackening after tumor exposure to magnetic hyperthermia, generated by iron oxide cubic nanocrystals [47]. This slackening allowed a better permeation of doxorubicin, which was applied intravenously in a combined treatment with magnetic hyperthermia [47]. Collagen fiber slackening was assessed post mortem and was characterized by a larger spacing between collagen fibers, probably due to the phase transition of the collagen, in tissues that underwent hyperthermia [47]. Conversely, in the case of photothermia, performed in this study, we could observe how photothermia kills cancer cells, or disrupts (or melts) the extracellular matrix in a 3D micro-tissue model. These phenomena were observed in real time after cells (Video S1) or cell-sheets (Video S2) exposure to the laser within a multi-photon microscope.

Photothermal treatment could have an enormous potential for tackling highly desmoplastic tumors, which have an extremely poor therapeutic outcome. The tumor microenvironment, particularly the tightly woven collagen matrix, has a recognized capability to reduce the penetration of conventional (molecular) therapeutics and immune cells, therefore the nanochains, which could disrupt the collagen matrix, have a significant therapeutic perspective. In the future, such nanochain-based platforms could offer entirely new, synergistic, and multimodal possibilities. The latter would combine physical and chemical means in one individual nanostructure platform, allowing magnetic guiding, drug delivery, and cellular matrix destruction, as well as tumor cell eradication upon heat activation with a source of light.

## 4. Materials and Methods

### 4.1. Nanoparticles Synthesis, Functionalization, and Characterization

#### 4.1.1. Raw Materials

The chemicals used for the synthesis of the superparamagnetic nanochains were of reagent grade quality and were obtained from commercial sources. Primary magnetic nanoparticle clusters were provided by Nanos SCI company (Ljubljana, Slovenia), and are commercially sold as iNANOvative™ (Nanos SCI, Ljubljana, Slovenia). Succinic anhydride (SA, 99%) and NH_4_OH (28–30%) were supplied by Alfa Aesar (Lancashire, UK). Acetone (AppliChem GmbH, Darmstadt, Germany) and ethanol absolute (Carlo Erba, reagent—USP, Milano, Italy) were used without further processing. Hydroxy (polyethyleneoxy) propyl triethoxysilane (silane-PEG, 50% in ethanol) was supplied by Gelest Inc. (Morrisville, PA, USA). Tetraethoxysilane (TEOS; 98%), rhodamine B isothiocyanate (RB), (3-aminopropyl) triethoxysilane (APS; silane-NH2, 99%), dichloromethane (DCM), dimethylformamide (DMF), Keiser test kit, and polyvinyl pyrrolidone (PVP, 40 kDa) were obtained from Sigma Aldrich (St. Louis, MO, USA).

#### 4.1.2. Syntheses of the Nanochains and Rhodamine-B-Labelled Nanochains

The commercial custom-made nanoparticle clusters with a ~3-nm-thick silica shell were provided by Nanos SCI company. The nanochain synthesis is based on dynamic magnetic assembly approach where parameters, such as (i) magnetic field strength, (ii) amount of TEOS added, (iii) duration of the exposure to the magnetic field, (iv) initial nanoparticle clusters concentration, and (v) nonmagnetic stirring rate, are precisely defined in order to control the nanochains length. The detailed syntheses procedures for the silica-coated nanoparticle clusters and nanochains have been published elsewhere [24,26,27,48]. Schematic demonstration of the crucial synthesis steps is presented in Appendix A. Briefly, nanochains composed of approximately 5 nanoparticle clusters were prepared as follows. The suspension of commercial nanoparticle clusters was admixed with the polyvinyl pyrrolidone (PVP, molecular weight 40 kDa, pH 4.3 adjusted by 0.1M HCl) solution to reach final PVP concentration of 1.25 × 10^−4^ M and the final nanoparticle cluster concentration of 2.2 × 10^−8^ M. The reaction mixture (volume 90 mL) was stirred mechanically with glass propeller at 250 rpm for 8.5 h. The nanochain fabrication took place at a TEOS concentration of 45 mM, and an exposure to a magnetic field of 65 ± 15 mT. The whole amount of TEOS was admixed into the reaction mixture 10 min after the transfer of the nanoparticle clusters into the PVP solution. The silica was deposited on the chain-like nanostructures formed by a magnetic assembly of nanoparticle clusters after the pH was increased to a value 8.5 using NH_4_OH. The pH was increased after 80 min of the TEOS addition. The nanochain synthesis with a ~15-nm-thick silica shell was completed in 3 h. The assemblies obtained by the mentioned procedure are denoted as “nanochains”. For fluorescent labelling, rhodamine-B (RB) was covalently integrated into the matrix of the silica shell. The reaction between RB and APS was carried out first in the mixture of DCM/DMF = 4/1 overnight at room temperature. RB (0.00933 mmol) was dissolved in the solvent mixture (0.5 mL) and then APS (0.186 mmol) was added. Subsequently, the volatile solvent was removed using nitrogen flow and the product (RB-APS) was mixed with TEOS for the fluorescent silica coating. RB-labelled nanochains are denoted as “RB-nanochains”. Finally, the synthesized nanochains and RB-nanochains were magnetically separated from the suspension and washed first with EtOH and then rinsed 3 times with distilled water.

#### 4.1.3. Nanochains Functionalization

In order to improve the colloidal stability, the nanochain surface was modified by functionalization with carboxyl-carrying moieties. First, the amino-functionalized nanochains were prepared with the grafting of APS to the nanochains and RB-nanochains silica surface, as described elsewhere [35,49]. In brief, nanochains (150 mg, 15 mL distilled water) were diluted with 15 mL of ethanol to which 75 µL of APS and 250 µL of NH_4_OH were added while the reaction mixture was stirred mechanically with glass propeller at 400 rpm for 5 h at 55 °C. The synthesized nanochains-NH_2_ were magnetically separated from the suspension and washed first with ethanol and then rinsed 2 times with distilled water and transferred in DMF. For the preparation of carboxyl-functionalized nanochains (nanochains-COOH), the nanochains-NH_2_ were further reacted with succinic anhydride (SA), applying a ring-opening elongation reaction. The nanochain-NH_2_ suspension (50 mg in 35 mL DMF) was led to react with SA (10 mg in 15 mL DMF) solution, where a 0.5 mL aliquot of the solution of SA in DMF was added per minute into nanochain-NH_2_ suspension, while being stirred mechanically with a glass propeller at 600 rpm overnight at room temperature. The produced nanochains-COOH formed a colloidal aqueous suspension. The same procedure of carboxyl functionalization was applied to RB-nanochains in order to produce RB-nanochains-COOH. Finally, both types of carboxyl-functionalized nanochains, nanochains-COOH and RB-nanochains-COOH, were thoroughly washed with acetone and ethanol, and dispersed in distilled water.

#### 4.1.4. Nanochains Characterization

The RB-nanochain-COOH structure was assessed by transmission electron microscopy (TEM). A drop of RB-nanochains-COOH suspension was deposited on a copper grid coated with a perforated, transparent carbon foil. The suspension was dried prior to TEM observations performed with a transmission electron microscope (Jeol, JEM, 2100, Akishima, Japan), operating at 200 kV. Magnetic properties of the RB-nanochains-COOH were measured at room temperature by vibrational sample magnetometry (VSM) (Lake Shore 7307 VSM). The zeta-potential measurements as a function of the pH of the RB-nanochains, RB-nanochains-NH_2_, and RB-nanochains-COOH suspensions (volume 15 mL) at final nanochains concentration of 0.2 mg/mL were monitored in an aqueous solution containing KCl (final concentration 10 mM). Zeta-potential measurements were performed on Zeta PALS, Brookhaven Instruments Corporation. FTIR-ATR analysis of the powders of the freeze-dried samples (RB-nanochains-NH_2_, and RB-nanochains-COOH; 15–20 mg each) was performed on Perkin Elmer, Spectrum 400 Spectrometer. The quantitative analysis of primary amines of the RB-nanochains-NH_2_ was determined by Keiser test, where accurately weighed 10 mg of the freeze-dried RB-nanochains-NH_2_ was applied in the reaction with ninhydrin while following the manufacturer’s protocol specified in the Kaiser kit.

### 4.2. Cell Culture and Loading of Cells with RB-Nanochains-COOH

The experiments were performed on four cell lines: the wild type human colorectal carcinoma cell line (HCT116) (ATCC^®^ CCL-247TM), green fluorescent protein-expressing human colorectal carcinoma cell line (HCT116–GFP) [50], HeLa cells (ATCC^®^ CCL-2TM) stably expressing GFP-RAB7 (HeLa GFP-Rab7) [51], and primary normal human dermal fibroblasts, isolated from a healthy skin biopsy [52]. Cells were grown the Dulbecco’s Modified Eagle Medium (DMEM, Gibco-Invitrogen, Carlsbad, CA, USA) containing 4.5 g/L glucose, L-Glutamine and pyruvate, 1% of penicillin/streptomycin, and 10% of fetal bovine serum (the medium with additives is denoted as cDMEM). The cells were grown under standard cell-growing conditions (5% CO_2_, 37 °C).

When the cells attained 60% confluence, they were rinsed with Dulbecco’s Phosphate Buffered Saline with calcium chloride and magnesium chloride (Gibco-Invitrogen; PBS in the following text) and were incubated for 1 h with silica-coated iron oxide nanochains, dispersed in RPMI Medium 1640 (which was not supplemented with glutamine, serum, or antibiotics). Cells were either incubated with rhodamine-loaded functionalized silica-coated iron oxide nanochains (RB-nanochains-COOH) at iron concentrations of 5 mM; or 5 mM rhodamine-free carboxyl functionalized silica-coated iron oxide nanochains (Nanochains-COOH). The nanochain suspension dilutions used for cell loading were obtained after diluting a volume of the concentrated stock suspension of RB-nanochains-COOH or Nanochains-COOH (which had a known concentration of 62 mM of iron, as determined by chemical and magnetic measurements) in the RPMI Medium 1640. Cell loading was performed in T-25 flasks containing 2 mL of RPMI and the aliquot containing nanochain suspension, to obtain the final concentration of 5 mM of iron. Nanochains-COOH were favored in experiments involving multiphoton imaging, described in Section 4.8, last paragraph. During this imaging protocol, we intended to avoid red-fluorescence signal of RB-nanochains-COOH, in order to visualize propidium iodide uptake, which emits in the same spectrum and was used in this case as a cell viability probe during laser exposure under the multiphoton microscope.

After the one-hour incubation period with nanoparticle suspension in RPMI medium at 37 °C, the cells were gently rinsed with PBS and placed in cDMEM for an overnight chase. Particle-loaded cells were detached following the standard trypsin/EDTA-based cell detachment protocol and either fixed and processed for transmission electron microscopy (TEM) analyses, or re-plated for bright field, fluorescence, and multiphoton imaging (which included laser exposure and heating induction) or used to form multicellular spheroids (MCS) in toxicity/proliferation tests.

Cell sheets were made with primary normal human dermal fibroblasts, as previously described [53]. In summary, thirty thousand cells were plated in 24-well plates containing a round filter paper band (approximately 3 mm wide), where they grew for 6 weeks. During this period, the cells were supplied with cDMEM freshly supplemented with 50 μg/mL ascorbic acid (Sigma- Aldrich, St Quentin Fallavier, France), three times per week. Cell sheets were rinsed with PBS and co-incubated with RB-nanochains-COOH at iron concentrations of 0.25 mM in 500 μL RPMI during an incubation period of 1 h at 37 °C. The sheets were then rinsed with PBS and cDMEM was added for the overnight chase.

### 4.3. Bright Field and Fluorescence Microscopy and Image Analysis

Cells were imaged with a wide field Leica DM IRB microscope (Leica Microsystems, Wetzlar, Germany) coupled to a CoolSNAP HQ camera (Roper Scientific, Photometrics, Tucson, AZ, USA), using the following filter sets: for the GFP λ_Ex_ = 480/40 nm and λ_Em_ = 527/30 nm, for the RB-nanochains-COOH λ_Ex_ = 560/40 nm and λ_Em_ = 630/75 nm, and for Hoechst stain λ_Ex_ = 360/40 and λ_Em_ > 425 nm.

Cellular spheroids were monitored for 9 days using the IncuCyte Live Cell Analysis System Microscope (Essen BioScience IncuCyte™, Herts, Welwyn Garden City, UK) at a magnification × 10.

Images were analyzed with the ImageJ software (U.S. National Institute of Health, Bethesda, MD, USA). The software was used to determine the mean fluorescence intensity of loaded cells, and to measure spheroid growth. The average fluorescence intensity was measured on the first day after cellular uptake of RB-nanochains-COOH and comprised all cells that were seeded to form a spheroid. The mean average intensity was obtained by averaging 6 intensities of cells forming 6 different spheroids per cell type. Spheroid growth follow-up was obtained by measuring spheroid diameters (as the spheroids formed sphere-like structures). From the diameter of the whole spheroid, measured on daily basis, we calculated the equatorial area, which was used for growth comparison.

### 4.4. Transmission Electron Microscopy of Biological Matter

Cells, cellular spheroids, and engineered cell sheets were fixed (2% glutaraldehyde in 0.1 M sodium cacodylate buffer), post-fixed (1% osmium tetroxide), gradually dehydrated in ethanol, and embedded (Embed 812 resin, Electron Microscopy Sciences). Thin sections (70 nm thick) were observed with a HT 7700 Hitachi transmission electron microscope equipped with a CCD AMT XR41 camera.

### 4.5. Nanochains Acute and Long-Term Toxicity Assessment

#### 4.5.1. Cell Viability Assessment by Trypan Blue Exclusion Test

Cell viability was determined 24 h after cellular uptake of RB-nanochains-COOH. The viability was assessed on the whole cell population: adherent cells and cells, which might have been floating in the culture medium. The media in which the cells were cultured and the tripsinized cells were pooled and centrifuged (5 min, 100× *g*). The supernatants were discarded and the pellets were re-suspended in PBS. Ten microliters of resulting cells suspensions were mixed with 10 µL of 0.4% Trypan Blue solution by gentle pipetting. Cells were manually counted using a Malassez hemocytometer within less than three minutes after preparation, and viability was calculated from the proportion of viable (undyed cells) in respect to the total number of cells (dyed and undyed). The counts were made in triplicates for each condition.

#### 4.5.2. RB-Nanochain-Loaded Cells Survival and Proliferation Assessment

A clonogenic assay [54] was used to assess the impact of the RB-nanochains-COOH on cell long term survival. Five hundred unloaded (control) or loaded cancerous cells (prepared as described in Section 4.2.) or 250 unloaded or loaded normal dermal fibroblasts were seeded in 6-well plates. Seven or fourteen days after seeding (in case of cancer cells or fibroblasts, respectively), the colonies were rinsed with PBS, fixed with methanol, and stained with crystal violet (0.5% *w/v*) for 15 min under gentle stirring at room temperature, rinsed 3 times with PBS, and left to dry prior colony counting.

The impact of RB-nanochains-COOH on cell survival and proliferation was also determined in 3D cell cultures, where chain-loaded cells were used to make cellular spheroids. Multicellular cell spheroids (MCS) were made following the non-adherent technique, by seeding cells in Costar^®^ Corning^®^ Ultra-low attachment 96-well plates (Fisher Scientific, Illkirch, France). Approximately five hundred unloaded or loaded cancer cells (HCT116, HCT116-GFP, or HeLa GFP-Rab7) of five thousand unloaded or loaded fibroblasts were seeded per well in 250 µL of cDMEM. Plates were cultivated in 5% CO_2_ humidified atmosphere at 37 °C. Single MCS were obtained in each well 24 h after seeding. The MCS growth was followed over a period of 9 days with video-microscopy, as described in Section 4.3. At the end of this period, the spheroids were fixed and processed for TEM, as described in Section 4.4.

### 4.6. Intracellular Iron Quantification by Single Cell Magnetophoresis

The iron load in loaded cells was determined by single cell magnetophoresis [55], which consists of measuring the velocity of cells loaded with magnetic nanoparticles, when loaded cells are submitted to a magnetic field gradient (B = 150 mT, gradB = 17 T/m). The migration of 100 cells was tracked by videomicroscopy for each cell-loading condition. Cell velocity and the average iron mass per cell were assessed as previously described [55].

### 4.7. Thermal Measurements

Thermal measurements were performed on nanoparticle aqueous suspensions. The measurements were made in 0.5 mL Eppendorf tubes containing 100 μL of RB-nanochains-COOH suspended in water, at an iron concentration of 130 mM. An alternating magnetic field generating device (DM3, NanoScale Biomagnetics, Zaragoza, Spain) operating at 471 kHz and 180 G was used to induce magnetic hyperthermia. Photothermal heating was obtained after NIR laser illumination operating at 808 nm at laser power densities of 0.3 and 1 W/cm^2^. The temperature elevation was measured as a function of time using an infrared camera (FLIR SC7000) and imaging the sample from the side.

### 4.8. Second Harmonic Generation Imaging and Photothermal Treatment of Plated Cells and Cell Sheets

Second harmonic generation (SHG) imaging was performed with a 7 MP multiphoton laser scanning microscope (Carl Zeiss, Jena, Germany), equipped with a mag. 20 × objective (with a numerical aperture of 0.95), coupled to a Ti–sapphire femtosecond laser, Chameleon Ultra 2 (Coherent Inc., Santa Clara, CA, USA) tuned to 880 nm. In order to avoid nanochain heating and obtain the image of engineered cell sheets, we fixed the laser power to 18%, attaining an equivalent of the laser power of 20 mW, while non-descanned detectors collected emitted light from nanochains, SHG, and dermal fibroblasts through 565–610 nm, 435–485 nm, and 500–550 nm bandpass filters, respectively. The micrographs were acquired using a laser dwell time of 0.58 or 0.92 µs per pixel at an X, Y resolution of 0.15 µm, and the micrographs were obtained after averaging 4 frames. In experiments which involved nanochains heating, the laser power was fixed to 26%, attaining an equivalent a the laser power of 28 mW, and 30 images were acquired over a period of 8 min (4 s/image). In experiments involving the heating of nanochain-loaded green fluorescent protein (GFP) expressing plated HCT-116 cells, the laser power was fixed at 40%, attaining an equivalent of a laser power of 33 mW, while non-descanned detectors collected emitted light from permeabilized cells (propidium iodide positive) and GFP through 565–610 nm and 500–550 nm bandpass filters, respectively. Sixty images were acquired during 2 min using a laser dwell time of 0.79 µs per pixel, at a X, Y resolution of 0.415 µm.

### 4.9. Statistical Analysis

All biological tests were made in triplicates performed in three independent experiments, except for experiences where spheroid growth was monitored over time, where 6 spheroids were made per condition and three independent experiments were made. The results are expressed as mean ± SEM and the differences between groups were assessed by unpaired Student *t*-test or two-way ANOVA depending on data set. A *p* value < 0.05 was considered significant.

## 5. Conclusions

In the present study, we characterized and biologically tested nanochains, prepared by magnetic assembly of nanoparticle clusters, and coated with an additional layer of fluorescent silica. These nanochains have an extraordinary therapeutic potential and are not toxic to different cancerous and non-cancerous cells (human dermal fibroblasts). After irradiation with near infrared light, such nanochains eradicate tumor cells in vitro and have the capacity to melt the collagen matrix, as showed using engineered cell sheets made of cells secreting their own extracellular matrix. Further tests, namely tests on large cell populations and animal studies, will now be undertaken to go beyond the proof of principle described in this study, and to ascertain the practical therapeutic value of presented nanochains. The capacity of a therapeutic agent to act concomitantly on cancer cells and their environment could be a game changer in cancer treatment.

## Figures and Tables

**Figure 1 cancers-11-02040-f001:**
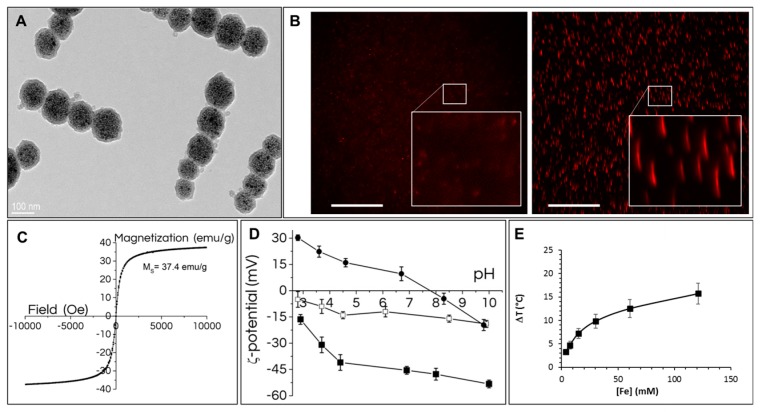
Structure and properties of RB-nanochains-COOH. (**A**) Transmission electron micrograph showing darker spherical nanoparticles cluster cores and less contrasted amorphous silica shell forming permanently sintered anisotropic nanochains, composed of ∼5 clusters per nanochain. (**B**) Fluorescence micrographs of RB-nanochains-COOH dispersed in Dulbecco’s Phosphate Buffered Saline (PBS) imaged (**left**) without the presence of a magnet and (**right**) with the magnet placed below the suspensions, showing the bundles of chains, which form well-defined, stable, transient super-assemblies, which align with the direction of magnetic field lines, and disassemble into individual nanochains as soon as the external magnet is removed. The inset shows a magnified view of the selected zone. Scale bar 100 µm (Mag. ×20). (**C**) Room-temperature measurements of the magnetization as a function of magnetic field strength (in emu/g of RB-nanochains-COOH). (**D**) The curves of the zeta-potential as a function of pH for RB-nanochains (white squares), for RB-nanochains-NH_2_ (black spheres) and RB-nanochains-COOH (black squares). (**E**) Temperature elevation curve of RB-nanochains-COOH in 100 µL of aqueous suspension measured in an Eppendorf tube upon laser irradiation at λ = 808 nm and laser power density of 0.3 W/cm^2^ at different iron concentrations expressed in millimoles (the bars represent the standard deviation (SD) obtained from 3 independent measurements).

**Figure 2 cancers-11-02040-f002:**
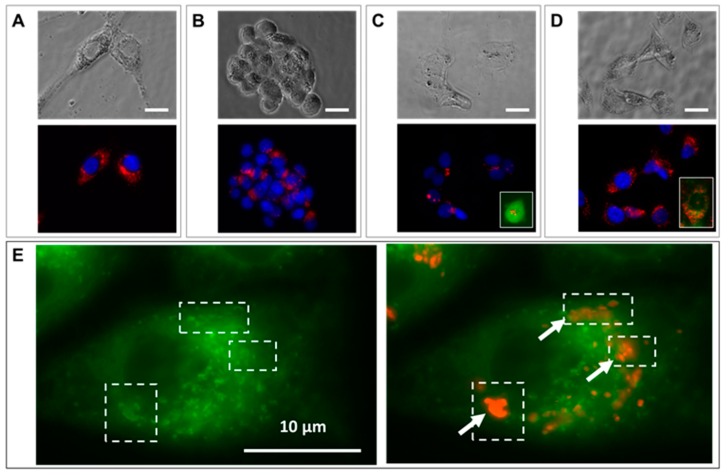
Cellular uptake of RB-nanochains-COOH obtained at an extracellular iron concentration of 5 mM within the RPMI medium. (**A**–**D**) Bright field (**top**) and fluorescence (**bottom**) micrographs showing RB-nanochain-COOH-loaded cells (**A**) Normal dermal fibroblasts, (**B**) HCT-116 wild type cells, (**C**) HCT-116 GFP cells, and (**D**) HeLa GFP-Rab7 cells (Scale bar 20 µm, Mag. ×40. Cell nuclei are stained in blue—Hoechst 33342 fluorescent stain. Insets in (**C**,**D**) show the intrinsic green fluorescence of GFP-expressing cells). (**E**) Left: Green fluorescence micrograph of a HeLa GFP-Rab7 cell, exhibiting characteristic green fluorescing endosomes, and right: green and red fluorescence micrographs overlay showing the co-localization of green fluorescing endosomes (white dashed line squares) with red fluorescing RB-nanochains-COOH clusters (squares and arrows) (Mag. ×100), (**F**) Representative transmission electron micrograph showing a loaded HCT-116 wild type cell (**left**) and the magnified view of internalized chains (**right**). N denotes the cell’s nucleus. (**G**) Iron quantification in loaded HCT-116 wild type cells, as determined by single cell magnetophoresis. The graph shows the distribution of internalized iron (expressed in picograms of iron per cell) after cells incubation with 5 mM extracellular iron concentration. (**H**) RB-nanochains-COOH internalization in different cell types expressed as mean fluorescence intensity determined in a population of 500 cells. The error bars represent the SD of mean fluorescence intensities determined from 6 experiments per cell type.

**Figure 3 cancers-11-02040-f003:**
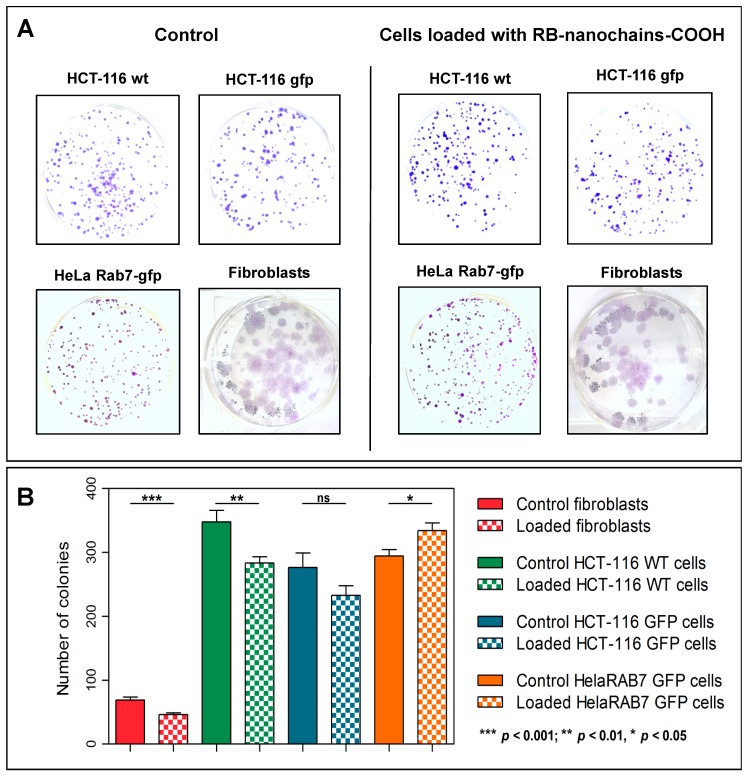
Colony-forming ability of control and RB-nanochain-COOH-loaded cells. (**A**) Images showing cell colonies after crystal violet staining. (**B**) Graph showing the number of colonies counted at D7 for cancer cell lines and at D 14 for normal dermal fibroblasts the term “loaded cells” refers to cells that internalized the RB-nanochains-COOH. The results (expressed as mean ± SEM) were obtained in three independent experiments made in triplicates. Differences between groups were assessed by an unpaired Student *t*-test and a *p*-value < 0.05 was considered significant.

**Figure 4 cancers-11-02040-f004:**
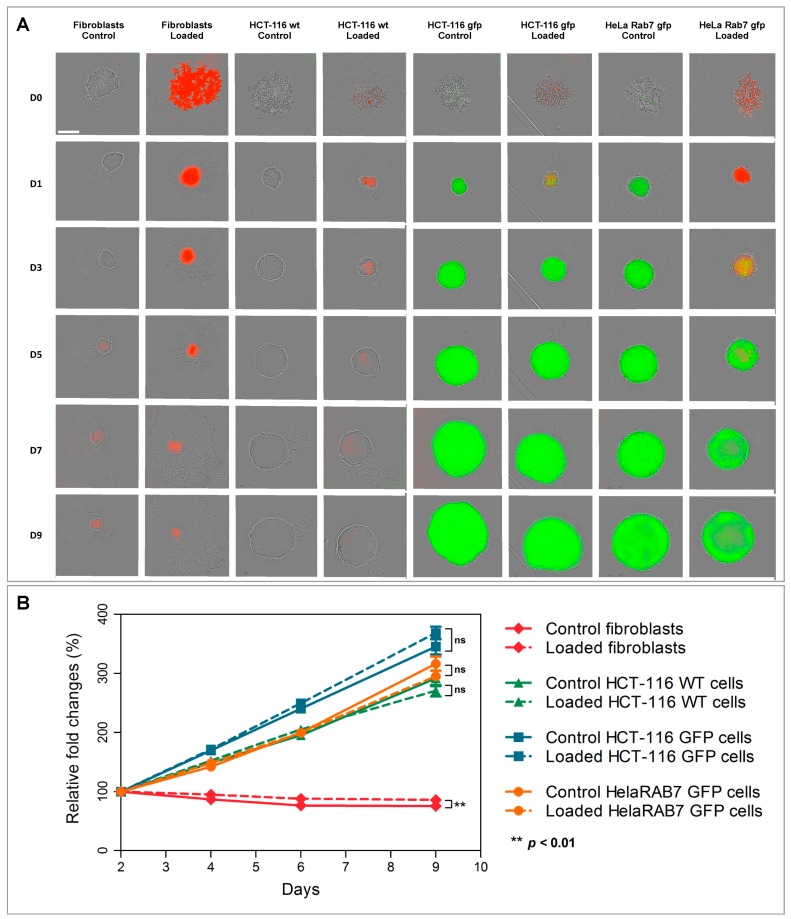
Control and RB-nanochain-COOH-containing cells (referred to as “loaded” in the figure), used to assess multicellular spheroids formation and growth. (**A**) Representative bright field and fluorescence micrographs overlays showing spheroids formation and growth over time. At the day of seeding (day 0—D0), the red fluorescence in cells, that internalized RB-nanochains-COOH, directly correlates with nanoparticle load. At D0 we see individual cells, which gradually agglomerate at day 1 after seeding (D1) and start forming cellular spheroids (D3 and onward). Scale bar 300 µm (mag. ×10). (**B**) Spheroid growth curve over time. Growth curves plotted from the area (mean ± standard error mean), *n* = 3 experiments, *n* = 6 spheroids per experiment per condition. Two-way ANOVA, *p*-value < 0.05 was considered significant.

**Figure 5 cancers-11-02040-f005:**
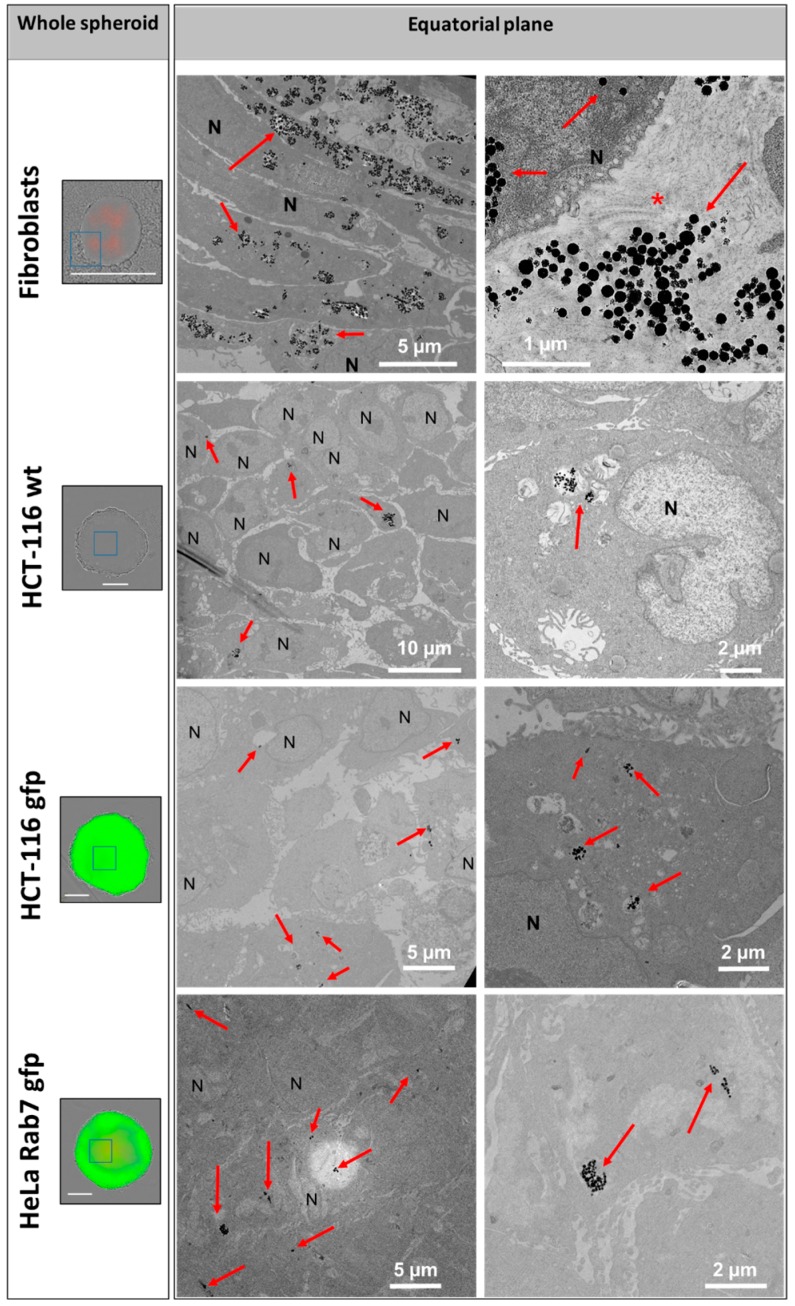
RB-nanochain-COOH distribution within multicellular spheroids. Left: Bright-field and fluorescence micrographs overlays of whole multicellular spheroids (scale bars 300 µm) fixed at day 11 and right: transmission electron microscopy (TEM) micrographs showing sections obtained at the equatorial plane of the spheroids. Nanochains are indicated with red arrows, N denotes cells’ nuclei, * denotes the collagen fibers as evidenced within the extracellular matrix of the fibroblasts in the top right panel.

**Figure 6 cancers-11-02040-f006:**
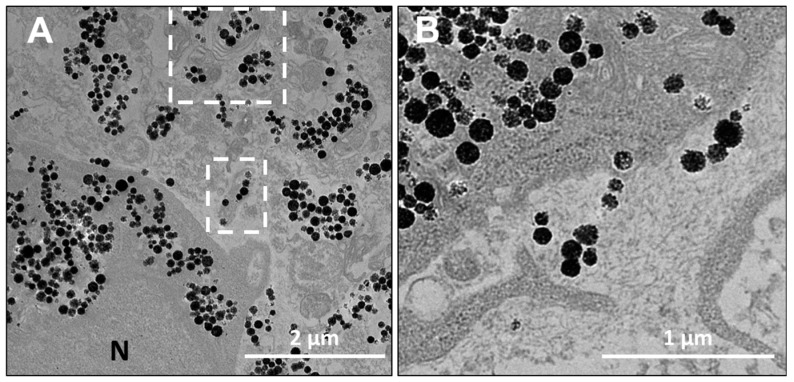
Comparison of extracellular locations of RB-nanochains-COOH within the extracellular matrix in multicellular spheroids. (**A**) Representative TEM micrograph showing RB-nanochains-COOH confined in extracellular vesicles (white dashed line squares) and (**B**) Non-confined RB-nanochains-COOH within fibroblast-secreted extracellular matrix.

**Figure 7 cancers-11-02040-f007:**
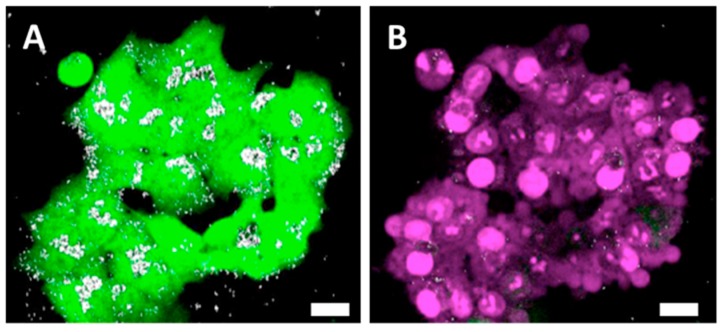
Nanochains as photothermal mediators affecting the cells and the extracellular matrix. (**A**) HCT-116-GFP cells (green) loaded with nanochains-COOH (white spots), exposed to the laser within the multiphoton microscope. (**B**) Nanochains-COOH-loaded HCT-116-GFP cells, which underwent cell death and thus internalized propidium iodide after laser exposure (the red fluorescence derived from cell-internalized propidium iodide is here represented by a magenta pseudo-color). (**C**) Representative micrograph of a control cell sheet exhibiting a rich collagenous matrix (turquoise) among auto-fluorescent (green) fibroblast cells (**D**) Representative micrograph of a cell sheet loaded with red-fluorescent RB-nanochains-COOH surrounded with collagen (turquoise) and fibroblasts (green). (**E**) TEM micrograph showing the distribution of RB-nanochains-COOH within the cell sheet. N denotes cell nucleus, yellow arrows point to intracellularly localized nanochains and red arrows point to extracellular nanochains. Green asterisks denote the collagen fibers in the extracellular matrix. (**F**) Representative micrograph of a cell sheet: exhibiting on the left side the intact collagen fibers (exposed to 20 mW laser power), and on the right side the melted collagen fibers (exposed to 33 mW laser power power). Scale bars in (**A**–**D**,**F**) equal 20 µm. White arrow in (**F**) points to a characteristic “drop” of melted collagen fibers.

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
