# Peer review of "Magnetic Silica-Coated Iron Oxide Nanochains as Photothermal Agents, Disrupting the Extracellular Matrix, and Eradicating Cancer Cells"

_cancers, 2019, doi:10.3390/cancers11122040_

Round 1

Reviewer 1 Report

In this paper, Kolosnjaj-Tabi et al. fabricated silica-coated iron oxide nanochains and applied their photothermal effect to cancer cell death. The nanoparticles were characterized and tested on tumor cells and spheroids. However, in my opinion, the overall strategy and data are not sufficient to demonstrate the novelty and key data and explanations are missing. Critical points are as below.

First of all, in high impact journals, most papers about nanomaterials for cancer therapy demonstrating their usefulness in animal studies. Therefore, only in vitro data that the nanochains have photothermal effect are not enough to show the excellence of this study.

Why the nanochains have photothermal effect? As the authors mentioned many times, iron oxide nanoparticles generally show thermal effect by magnetic field. The authors did not provide sufficient explanation about it.

Why the nanochains have chain structure? It is intended? Is there any advantages in chain structure?

No error bars in Fig 1E and 2G.

No scale bar in Fig 2A, 2B, 2C, 2D, and 2E.

A graph showing quantitative analysis of cellular uptake is needed in Fig 2.

It is hard to observe co-localization of particles and endosomes in Fig 2E.

In Fig 4, why did the size of spheroid decrease from D0 to D1?

There is no quantitative analysis about photothermal cell death and ECM destruction. These data are essential.

The authors wrote ‘In cellular spheroids composed of normal dermal fibroblasts, the nanochains are localized in endo-lysosomal compartments within the cells, and within the extracellular matrix, while in MCS made of cancer cells the nanochains are exclusively found in the endo-lysosomal compartments within cells.’ What is the reason? More particles were internalized in fibroblast in Fig 2. It is not reasonable.

Author Response

We thank the reviewers for their constructive comments and provide answers to all questions they raised.

Reviewer 1:

Q1: First of all, in high impact journals, most papers about nanomaterials for cancer therapy demonstrating their usefulness in animal studies. Therefore, only in vitro data that the nanochains have photothermal effect are not enough to show the excellence of this study.

A1: While we understand the reviewer’s point of view, a large number of studies is not in agreement with this statement.  For informational purposes to the reviewer, we here list just some of the studies published in high impact journals, that do not involve animal studies. The publications of the latest issues of several high impact journals, involves assessing the effect of nanoparticles, molecular medicines, prospective therapeutic agents, and even mechanisms involved in cancer treatment and cancer response.

Published in the Special Issue "Cancer Nanomedicine" in the MDPI Cancers journal:

Santos-Rebelo, Ana, et al. "Development and Mechanistic Insight into the Enhanced Cytotoxic Potential of Parvifloron D Albumin Nanoparticles in EGFR-Overexpressing Pancreatic Cancer Cells." Cancers11 (2019): 1733. Sambi, Manpreet, et al. "Next-Generation Multimodality of Nanomedicine Therapy: Size and Structure Dependence of Folic Acid Conjugated Copolymers Actively Target Cancer Cells in Disabling Cell Division and Inducing Apoptosis." Cancers11 (2019): 1698 Nieto, Celia, et al. "Size Matters in the Cytotoxicity of Polydopamine Nanoparticles in Different Types of Tumors." Cancers11 (2019): 1679. Hasan, Md, et al. "Multi-Drug/Gene NASH Therapy Delivery and Selective Hyperspectral NIR Imaging Using Chirality-Sorted Single-Walled Carbon Nanotubes." Cancers8 (2019): 1175. Latorre, Ana, et al. "Multifunctional albumin-stabilized gold nanoclusters for the reduction of cancer stem cells." Cancers7 (2019): 969. Bugárová, Nikola, et al. "A Multifunctional Graphene Oxide Platform for Targeting Cancer." Cancers6 (2019): 753.

Published elsewhere:

Nashimoto, Yuji, et al. "Vascularized cancer on a chip: The effect of perfusion on growth and drug delivery of tumor spheroid." Biomaterials229 (2020): 119547. Mandal, Kalpana, et al. "Role of a kinesin motor in cancer cell mechanics." Nano Letters11 (2019): 7691-7702. Wang, Jianhong, et al. "Core-shell tecto dendrimers formed via host-guest supramolecular assembly as a pH-responsive intelligent carrier for enhanced anticancer drug delivery." Nanoscale(2019).

Q2: Why the nanochains have photothermal effect? As the authors mentioned many times, iron oxide nanoparticles generally show thermal effect by magnetic field. The authors did not provide sufficient explanation about it.

A2: Iron oxides (therefore also the nanochains used in this study) absorb light, and the energy of absorbed light is transformed to heat. This explanation was given in line 53 of our manuscript and further explanations are now added in the amended text (please refer to paragraph 7 of the introduction section). The reviewer is also right that for a long time, magnetic hyperthermia has been considered as the only nanoparticles-mediated thermal modality for cancer treatment. However, few years ago, their remarkable conversion of light to heat in the near infrared window has been demonstrated. Since then, photothermia is more and more preferred to magnetic hyperthermia for heat generation. Moreover, photothermia was found way more efficient then magnetic hyperhtermia, as we (see our reference Espinosa, Adv Funct Mat 2018) and others have recently evidenced. The efficiency of photothermia especially occurs when nanoparticles are confined within endosomes, because then, the magnetic hyperthermia heating is strongly hampered. The drawbacks of magnetic hyperthermia are mentioned in the introduction and further discussed in discussion section.

Q3: Why the nanochains have chain structure? It is intended? Is there any advantages in chain structure?

A3: We thank the reviewer for these questions. In order to be useful in biomedicine, nanoparticles should be superparamagnetic. Superparamagnetism is a form of magnetism where magnetization randomly flips at room temperature. In the absence of an external magnetic field, the overall magnetization of a group of superparamagnetic nanoparticles (smaller than 20 nm for iron oxides) is zero because the magnetic moment of the nanoparticles is randomly distributed. The weakness of the attractive magnetic interactions among the superparamagnetic nanoparticles do not to allow them to magnetically aggregate. This magnetic behavior is crucial for the preparation of ferrofluids, which can therefore be applied to biological systems, because superparamagnetic nanoparticles remain well dispersed. Nevertheless, when an external magnet is applied, all magnetic moments are aligned in the same direction, leading to a net magnetization. Unfortunately, individual superparamagnetic nanoparticles are too small for their effective translational movement even in the strongest magnetic fields. The only solution for preserving superparamagnetism and achieving strong magnetic responsiveness is to group a number of small superparamagnetic nanoparticles into larger and defined nanoparticle clusters. Such clusters are highly magnetically responsive and therefore form chain-like structures once exposed to magnetic field. This is the basis for obtaining our permanent chain-like particles, i.e., nanochains. The nanochains where obtained when a suspension of superparamagnetic nanoparticle clusters was exposed to defined magnetic field, while silica was added, to has fixated the clusters into permanent nanochains. Such nanochains are superparamagnetic with superior magnetic responsiveness due to much larger magnetic moment (larger magnetic volume) than the one of individual nanoparticle clusters. We have added these information into the introduction.

Finally, the obtained nanochains were well designed, prepared intentionally, biologically tested and represent promising and very unique nanomaterial possessing superparamagnetism, superior magnetic responsiveness (see Figure S1 in Supplemental Information), and good photothermal properties which have been rarely combined in single nanostructures.

No error bars in Fig 1E and 2G.

We thank the reviewer for this comment. Figure 1E is now provided with the error bars corresponding to the measurements. We apologize for omitting them initially.

Figure 2G represents the distribution of the cellular uptake by cells. The measurements were retrieved after analysis of 100 cells. Each bar represents a population of cells, which internalized the amount of iron noted on the y axis. Such graphs do not include error bars. The mean +/- sd (17.3 +/- 2.6) are written in the figure and in the text. Besides, we have now added fluorescence measurements as an additional way to determine uptake, with graph showing the mean fluorescence intensity, averaged over different cellular populations.

No scale bar in Fig 2A, 2B, 2C, 2D, and 2E.

In micrograph panels, it is common to provide scale bars or give the magnification used or provide both. As the panels are small (figure A-D), we previously chosen to provide the magnification details in the caption of the figure. The same was applied to figure E in order not to overwrite the presented cell. Nevertheless, as the reviewer requested, we now added both, the magnifications and the scales.

A graph showing quantitative analysis of cellular uptake is needed in Fig 2.

We have now added the quantitative uptake extracted from the mean cellular fluorescence intensity, reflecting the amount of internalized nanochains.

It is hard to observe co-localization of particles and endosomes in Fig 2E.

We have added squares and arrows to facilitate the observations.

Q4: In Fig 4, why did the size of spheroid decrease from D0 to D1?

A4: The spheroids were made following the non-adherent technique, by seeding cells in ultra low attachment plates. This technique consists in placing cells into wells, as seen on D0. At day 0 after cells seeding we can observe individual cells within the well. As the cells are repelled by the coating of the wells and attracted by gravity, the cells gradually aggregate together (D1). Gradually, typically at D3, the cells, which get close to each other, start forming cell-to-cell communications channels, including gap junctions, desmosomes and electrical coupling, and their own extracellular matrix. Subsequently and over time, the spheroids start featuring a proliferative, a quiescent and a necrotic layer, mimicking the avascular region of in vivo tumor tissues. At D0 we thus observe cells and from D0 to D3 we spot the formation of the spheroid, so at the beginning we see the cells which are unconstrained and are gradually “forced” to form spheroids, in which cells come tightly together and thus decrease their apparent size. We have added the phrase: “At the day of seeding (day zero- D0) we see individual cells, which gradually agglomerate at day 1 after seeding (D1) and start forming cellular spheroids (D3 and onward)” in figure 4 caption.

Q5: There is no quantitative analysis about photothermal cell death and ECM destruction. These data are essential.

A5: As stated in the text, all cells, which internalized nanochains, underwent cell death, irrespective of the cell type. The cell death was global (reaching 100%). The outcome is shown in figure 6 and in the supporting information (SI). The ECM destruction was also observed wherever the nanoparticles were present, as evidenced in figure 6 and in the video in SI.

Q6: The authors wrote ‘In cellular spheroids composed of normal dermal fibroblasts, the nanochains are localized in endo-lysosomal compartments within the cells, and within the extracellular matrix, while in MCS made of cancer cells the nanochains are exclusively found in the endo-lysosomal compartments within cells.’ What is the reason? More particles were internalized in fibroblast in Fig 2. It is not reasonable.

A6: Cancer cells divide and split the nanoparticulate load much faster than normal cells. In contrast, fibroblasts doubling time is greater and they exhibit contact inhibition of proliferation, which translates to spheroids slower growth. As we state in the manuscript, the nanoparticle burden per cell is therefore greater in normal cells-derived spheroids.

Previous studies showed that the gene expression of ferritin, the endogenous iron storage protein, and other proteins involved in iron homeostasis, are upregulated after cell internalization of iron oxide nanoparticles (E. Pawelczyket al., NMR Biomed. 2006, 19, 581; N. Feliu et al., Chem. Soc. Rev. 2016, 45, 2440.). Moreover, ferritin expression is upregulated in different cancers, as numerous studies suggest (Several research articles and reviews are available to date, to give you just one example please see the review recently published in Cancers MDPI: M.S. Petronek et al., 2019 Jul 30;11(8). pii: E1077. doi: 10.3390/cancers11081077). This is probably the reason why the colony forming ability of cancer cells was less affected.

In contrast, normal cells, which divide slowly, and do not exhibit an upregulated expression of ferritin, accumulate the nanoparticles within cellular compartments. Nevertheless, these nanoparticles can be “expelled” by cells by extracellular vesicles.  This is why nanochains can be found within the extracellular matrix. Based on this evidence (we have now added an additional figure in the text), we stipulate that normal cells (= which internalize more nanoparticles, have a smaller proliferation rate and a higher death rate in respect to the global number of cells constituting the spheroids), generate more extracellular vesicles, which delocalize nanoparticles outside the cells into the extracellular matrix.

Reviewer 2 Report

The manuscript describe a class of silica coated iron oxide nanochains that be used as photothermal agents to facilitate hyperthermia. The nanochains were characterized for their charge properties, magnetic properties, morphology, and photothermal properties. The work utilizes 2D and 3D cell cultures models to evaluate the photothermal heating via the nanochains. In addition to causing tumor cells death, the heating also melts the collagen matrix. Therefore, the nanochains can potentially be utilize as the photothermal agents to facilitate hyperthermia.

General comments

The manuscript needs to provide stronger rationale to justify the decision for using nanochain. The cited literatures mentioned about the potential of using IONP-induced photothermia. This only supports the decision about using IONP. Choosing nanochain also needs to be clearly justified. Otherwise, nanochain is just a random nanomaterial. The manuscript states the nanochain has good colloidal stability. However, there is no to data to support it. The manuscript needs to clearly describe the mechanism of photothermal heating associated with the nanochain. The manuscript needs to quantitate the amount of internalize iron, which is needed to support the discussion about the toxicity. Only showing the concentration during cell treatment cannot be used to estimate the uptake. The manuscript needs to provide stronger justification for the utilized approaches to evaluate toxicity. Cell viability and toxicity are commonly evaluated using various standard colorimetric or fluorescent assays, which are more common and can be understood by the audiences. Does the utilized approaches exhibit any advantages over the common assays? The manuscript needs to provide more information about the nanochain synthesis. For example, how to control the nanochain to contain mainly 5 clusters? The manuscript needs to be carefully edited. For example, line 79 is like a random sentence insert at the end of introduction. There are also few paragraphs only has one sentence.

Specific comments

Line 96, the manuscript states the evaluation of the functionalization efficiency using zeta-potential. However, the data only show electrophoretic mobility versus pH, which doesn’t demonstrate the “efficiency” of functionalization.

Line 97, the manuscript shows the results of silica-coated nanochain and carboxyl functionalized nanochain, which are quite confusing. The manuscript should clearly state the difference between two nanochains. My understanding is all nanochains have silica coating.

Line 107, the manuscript uses zeta-potential data to characterize surface functionalization. However, zeta-potential data only demonstrate the charge property. The functionalization needs to be characterized using other surface chemistry characterization approaches such as FTIR, XPS, etc.

Line 144, the manuscript needs to use elemental analysis to quantitate the amount of internalize iron. Magnetophoretic mobility can only be used to characterize the movement under an applied magnetic field.

Line 234, the manuscript needs to explain why the surfaces with carboxylates would enable better cellular uptake. The cell surfaces are negatively charged, which is repulsive.

Line 239, the manuscript needs to clearly describe how data in Figure 4 demonstrate different amount of internalized nanoparticles.

Line 258, the discussion about the pro-proliferative effect is very confusing and contradicting to the discussion about the higher uptake amount for fibroblasts.

Line 377, the amino-surface is also charge and the carboxylate functionalization is utilizing the surface amine groups. There is no obvious reason why carboxyl nanochain is more colloidal stable.

Minor comments

Line 134, the manuscript should consider a different word choice. The phenomenon described in the manuscript is a cellular uptake, NOT labeling.

Figure 4, the labels D0-D9 need clear description.

Line 316, the reference is missing.

Author Response

We thank the reviewer for the questions and comments that helped improve the manuscript. Here are our answers

Q1: The manuscript needs to provide stronger rationale to justify the decision for using nanochain. The cited literatures mentioned about the potential of using IONP-induced photothermia. This only supports the decision about using IONP. Choosing nanochain also needs to be clearly justified. Otherwise, nanochain is just a random nanomaterial.

A1: We thank the reviewer for the comment. One of the very important advantages of the magnetic nanochains is their superior magnetic responsiveness, which could be used for all applications where magnetic guidance is desired, such as magnetic drug delivery or magnetic manipulation of targeted nanostructures. We added this explanation in the introduction section and provided an additional Figure S1 in Supplemental Information to demonstrate the efficacy of magnetic responsiveness. We have also provided better background that justifies the advantage of magnetic properties of chains over individual magnetic nanoparticles.

Q2: The manuscript states the nanochain has good colloidal stability. However, there is no to data to support it.

A2: We thank to the reviewer for the comment. There are no very direct methods for quantitative assessment of colloidal stability. However, dynamic light scattering is frequently applied for determination of colloids hydrodynamic size which is, unfortunately, designed for isotropic (spherical) particles. The method is therefore useless for characterization of nanochains due to their highly anisotropic shape. Alternatively, the behavior of colloidal stability in suspension could be estimated by monitoring the spontaneous sedimentation of the colloids over time. The RB-nanochains-COOH remained in suspension (cellular medium) at least three weeks and then only gradual color change could be seen indicating good stability for such relatively large “colloids”. The RB-nanochains-COOH sedimented completely due to gravity in two months but they could be easily redispersed with single gentle shake by hand. Another indirect method is the measurement of the RB-nanochains-COOH zeta potential where high absolute values (over ±30mV) at physiologically relevant pH 7.4 announce good colloidal stability. In contrary, RB-nanochains are slightly negatively charged (-12 ± 3 mV) while RB-nanochains-NH2 are slightly positively charged (6 ± 5 mV) at pH 7.4 which are both too low absolute values for achieving suitable colloidal stability for biological settings. We therefore applied carboxyl-functionalized nanochains (nanochains-COOH and RB-nanochains-COOH) for our studies. These explanations are now added to the Discussion section.

Q3: The manuscript needs to clearly describe the mechanism of photothermal heating associated with the nanochain.

A3: Matter can absorb, transmit or reflect light. The absorbed radiation energy of light can be converted to heat, while the reflected or transmitted light are scattered in space. The heating, which occurs when light interacts with nanomaterials, can have different physical mechanisms. In metals, such as gold, the light to heat conversion occurs when light interacts with conduction electrons on the surface of metallic nanoparticles. In carbon-based nanoparticles, the delocalized electrons absorb light and the energy is converted to vibrations of the C-C reticule, which is released as heat when the vibrational states decay. In other semiconductors, such as iron oxides or quantum dots, the band gap between valence and conduction electrons is small enough, to be bridged by excitation, provided by the optical energy (the photons) of a laser beam. After excitation, valence electrons stay in the conduction band only for a limited time, after which they fall back to the valence position. When this fall occurs, the electromagnetic radiation is emitted (Sattler, Klaus D. Handbook of nanophysics: nanomedicine and nanorobotics. CRC press, 2010). In the same way as the band gap has a specific value, which is characteristic for a specific type of material, the dissipated energy (heat) or the emission frequency (light), which occur when the electrons fall back to the valence level, have a fixed specific value. This explication was now added to the text.

The photothermal action of nanochains thus emerges due to intrinsic properties (=light absorption) of iron oxides.

Q4: The manuscript needs to quantitate the amount of internalize iron, which is needed to support the discussion about the toxicity. Only showing the concentration during cell treatment cannot be used to estimate the uptake.

A5: We agree with the reviewer. The iron concentration in the cellular medium was provided mainly for informative purposes, but the internalization of nanoparticles was quantified with magnetophoresis and fluorescence measurements (Figure 2 G and H). As rhodamine is covalently bound within the silica shell and does not undergo bleaching, the fluorescence is stable and correlates with nanochains concentration

Q6: The manuscript needs to provide stronger justification for the utilized approaches to evaluate toxicity. Cell viability and toxicity are commonly evaluated using various standard colorimetric or fluorescent assays, which are more common and can be understood by the audiences. Does the utilized approaches exhibit any advantages over the common assays?

A6: We thank the reviewer for raising these points. The reviewer is right that cell viability (and thus toxicity) is commonly evaluated using metabolic assays, which rely on read-out principles that include absorbance (colorimetry), fluorescence or luminescence. Nevertheless, as iron oxides absorb light, colorimetric assays (such as MTT or MTS) or bio-luminescent (such as the ATP-based assays) might lead to false readouts, as studies reported. Moreover, iron oxides can act as fluorescence quenchers, therefore redox-based viability tests, using fluorescent dyes, were also considered inappropriate for our study. We therefore assessed the viability with the trypan blue exclusion test. This test relies on the capacity of viable cells to exclude the dye and the viability was assessed by direct visualization of cells and manual counting. As trypan blue exclusion only provides information about early events that occur after nanoparticles uptake, we have used 2 additional tests to evaluate the long term effects. These tests include the clonogenic assay and the growth of spheroids made with nanoparticles that internalized nanochains. These two additional tests provided evidence that cells are not affected by nanoparticles even on long term. We have now added this text and additional relevant references.

Q7: The manuscript needs to provide more information about the nanochain synthesis. For example, how to control the nanochain to contain mainly 5 clusters?

We thank the reviewer for the comment. We have now inserted into the manuscript all the information needed to reproduce the synthesis. The nanochains synthesis is based on dynamic magnetic assembly approach where good control over parameters, such as i) magnetic field strength, ii) amount of TEOS added, iii) duration of the exposure to the magnetic field, iv) initial clusters concentration, and v) nonmagnetic stirring rate, assure the fabrication of the nanochains with defined length. Due to the reaction complexity the simultaneous control over all the parameters is very challenging and the material is therefore unique. We have now edited in more details the nanochains synthesis part in the main text and provided additional information.

Q8: The manuscript needs to be carefully edited. For example, line 79 is like a random sentence insert at the end of introduction. There are also few paragraphs only has one sentence.

We have changed the end of introduction, which now states “Cancerous cells and their microenvironment both play a pivotal role in cancer development, progression and resistance to treatment. In this regard, the nanochains, presented herein, which heat and thus simultaneously affect cellular and environmental components, could radically improve the therapeutic outcome.

Specific comments

Line 96, the manuscript states the evaluation of the functionalization efficiency using zeta-potential. However, the data only show electrophoretic mobility versus pH, which doesn’t demonstrate the “efficiency” of functionalization.

A: We totally agree with the reviewer’ comment. We quantified the primary amines of the RB-nanochains-NH2 by Keiser test (commercial “Keiser test kit” from Sigma Aldrich) following the supplied protocol. The test is based on the reaction of ninhydrin with primary amines, which results in the formation of dark blue color. The spectroscopically determined mean value (in triplicates) of accessible amines on the surface was 145 µmol per gram of the RB-nanochains-NH2.

Furthermore, a FTIR-ATR surface characterization analysis was additionally performed in order to confirm the successful reaction of surface amines of the RB-nanochains-NH2 with SA forming free –COOH groups on the surface of the RB-nanochains-COOH. The FTIR-ATR spectra are included as Figure S2 in Supplemental Information.

Finally, we performed an additional zeta potential measurement, although the method is not suitable to confirm the efficiency, of the RB-nanochains-NH2 in order to demonstrate different surface behavior (positive surface charge) at pH 7.4 compared to the one of the RB-nanochains-COOH (negative surface charge) which were prepared from the RB-nanochains-NH2 confirming the success in binding of SA to the RB-nanochains-NH2. The additional curve of the zeta potential of the RB-nanochains-NH2 was inserted in Figure 1D.

Line 97, the manuscript shows the results of silica-coated nanochain and carboxyl functionalized nanochain, which are quite confusing. The manuscript should clearly state the difference between two nanochains. My understanding is all nanochains have silica coating.

A: We thank to the reviewer for the comment. We carefully edited the entire manuscript in order to avoid any further misunderstanding. We believe the manuscript is significantly improved in terms of precision and clarity.

Yes, the nanochains cannot be fabricated at all if magnetically assembled nanoparticle clusters would not be fixated with a layer of silica to link them together permanently.

Line 107, the manuscript uses zeta-potential data to characterize surface functionalization. However, zeta-potential data only demonstrate the charge property. The functionalization needs to be characterized using other surface chemistry characterization approaches such as FTIR, XPS, etc.

A: We agree with the reviewer’ comment. We quantified the primary amines of the RB-nanochains-NH2 by Keiser test (commercial “Keiser test kit” from Sigma Aldrich) following the supplied protocol. The test is based on the reaction of ninhydrin with primary amines, which results in the formation of dark blue color. The spectroscopically determined mean value (in triplicates) of accessible amines on the surface was 145 µmol per gram of the RB-nanochains-NH2.

Furthermore, a FTIR-ATR surface characterization analysis was additionally performed in order to confirm the successful reaction of surface amines of the RB-nanochains-NH2 with SA forming free –COOH groups on the surface of the RB-nanochains-COOH. The FTIR-ATR spectra are included as Figure S2 in Supplemental Information.

Finally, we performed an additional zeta potential measurement, although the method is not suitable to confirm the efficiency, of the RB-nanochains-NH2 in order to demonstrate different surface behavior (positive surface charge) at pH 7.4 compared to the one of the RB-nanochains-COOH (negative surface charge) which were prepared from the RB-nanochains-NH2 suggesting the success in binding of SA to the RB-nanochains-NH2. The additional curve of the zeta potential of the RB-nanochains-NH2 was inserted in Figure 1D.

Line 144, the manuscript needs to use elemental analysis to quantitate the amount of internalize iron. Magnetophoretic mobility can only be used to characterize the movement under an applied magnetic field.

In this point we respectfully disagree with the reviewer. In the single cell magnetophoresis, which we applied, we determined the magnetically induced velocity of a number (minimally equaling 100) magnetically labeled cells in a magnetic field gradient. The analogy of cell magnetophoresis with the well-known cell electrophoresis must be considered with caution: contrary to electrophoresis, where charge movement is driven by a uniform electric field, a magnetic moment can experience a magnetic force only in the presence of a magnetic field gradient dB. Magnetically labeled cells bearing a global magnetic moment M are thus attracted in the region of the maximum magnetic field. As the carrier medium for the cells is non-magnetic, the magnetic force that puts magnetically labeled cells in suspension into motion is:

Fm =MxgradB

where B is the magnetic field experienced by the cell. The total magnetic moment of the cell, M is simply the product of the effective magnetization (meff), of one particle in the field B by the total number N of particles withinthe cell: M= Nmeff. Cells moving in a liquid medium are submitted to a hydrodynamic drag force, Fd. At low Reynolds number, assuming a perfect sphere for the cell, Fd is given by Stokes law: Fd=-3πηDv, where e D is the cell diameter, η is the viscosity of the carrier liquid and v is the cell velocity. In a permanent regime, Fd counterbalances the magnetic force so that the velocity measurement of each cell leads to the cell magnetization and particle load N.

This method was previously validated (Wilhelm, C.; Gazeau, F.; Bacri, J.-C. Magnetophoresis and ferromagnetic resonance of magnetically labeled cells. European Biophysics Journal 2002, 31, 118-125) and is a robust tool to quantify magnetic nanoparticles uptake.

Line 234, the manuscript needs to explain why the surfaces with carboxylates would enable better cellular uptake. The cell surfaces are negatively charged, which is repulsive.

Indeed, cationic surface charge generally correlates with a higher cellular uptake, yet, positively charged nanoparticles have a greater toxicity, at least in non phagocytic cells. Cationic NPs either directly or indirectly (by detachment of adsorbed polymers, such as polyethylenimine) cause membrane damage (Fröhlich, Eleonore. "The role of surface charge in cellular uptake and cytotoxicity of medical nanoparticles." International journal of nanomedicine 7 (2012): 5577.). This is why anionic nanoparticles are generally favored over cationic nanoparticles.

As the reviewer pointed out, the membrane is negatively charged. Nevertheless, the culturing medium is a complex (ionic) environment (One liter of RPMI 1640 contains: Glucose (2 g) pH indicator (phenol red, 5 mg) Salts (6 g sodium chloride, 2 g sodium bicarbonate, 1.512 g disodium phosphate, 400 mg potassium chloride, 100 mg magnesium sulfate, and 100 mg calcium nitrate). The medium thus consists of a pool of cations that can be adsorbed on positively-charged nanoparticles, prior to their interaction with the cell membrane surface. Here we emphasize that our “labelling” procedure consists of incubating the cells in serum free culturing medium (the RPMI). Cell loading with anionic magnetic nanoparticles was proven as simple, fast, efficient and universal. Moreover, when anionic nanoparticles are internalized the cell preserves its functionalities, as previously shown (Wilhelm, Claire, and Florence Gazeau. "Universal cell labelling with anionic magnetic nanoparticles." Biomaterials 29.22 (2008): 3161-3174.).

Line 239, the manuscript needs to clearly describe how data in Figure 4 demonstrate different amount of internalized nanoparticles.

We thank the reviewer for the remark. Different amounts of internalized nanoparticles in growing spheroids were obtained by the analysis of intensity of fluorescence, which correlates with the amount of internalized fluorescent nanochains. Here we point out that the rhodamine is bound within the silica shell, and is not on the surface, therefore can not be detached from the surface. The fluorescence yield thus directly correlates with the amount of internalized nanochains.

Line 258, the discussion about the pro-proliferative effect is very confusing and contradicting to the discussion about the higher uptake amount for fibroblasts.

The pro-proliferative effect as well as nanoparticles impact on cells may be cell-type dependent, as studies suggest (e.g. Abdal Dayem, Ahmed, Soo Bin Lee, and Ssang-Goo Cho. "The impact of metallic nanoparticles on stem cell proliferation and differentiation." Nanomaterials 8.10 (2018): 761.). Nevertheless, we are thankful to the reviewer for rising this point, because it made us reconsider our wording. The results of the clonogenic assay provide the result on the colony forming ability of the cells, which does not necessarily correlate with a pro-proliferative effect. As in the spheroid growth follow up the spheroids did not grow faster after nanoparticle internalization, we can not talk about a pro-proliferative effect. Therefore we amended this in the text.

Line 377, the amino-surface is also charge and the carboxylate functionalization is utilizing the surface amine groups. There is no obvious reason why carboxyl nanochain is more colloidal stable.

A: We thank to the reviewer for the comment. We performed an additional characterization. There are no very direct methods for quantitative assessment of colloidal stability. However, zeta potential, as an indirect method, of the RB-nanochains-COOH shows high absolute values (over ±30 mV) at physiologically relevant pH values reflecting good colloidal stability. In contrary, RB-nanochains are slightly negatively charged (-12 ± 3 mV) while RB-nanochains-NH2 are slightly positively charged (6 ± 5 mV) at pH 7.4 which are both too low absolute values for achieving suitable colloidal stability for biological settings. We therefore applied carboxyl-functionalized nanochains (nanochains-COOH and RB-nanochains-COOH) for our studies.

Line 134, the manuscript should consider a different word choice. The phenomenon described in the manuscript is a cellular uptake, NOT labeling.

We amended the wording. Instead of cell labeling we now use cellular uptake or cellular loading with nanochains.

Figure 4, the labels D0-D9 need clear description.

We thank the reviewer for the comment. The description was added in the figure caption.

Line 316, the reference is missing.

We thank the reviewer for the comment. The reference was added ( Shen, Shun, et al. "Magnetic nanoparticle clusters for photothermal therapy with near-infrared irradiation." Biomaterials 39 (2015): 67-74.)

Reviewer 3 Report

This is a well done research, but some experiments need to be completed.

1.Please show the amount of COOH ligands modified on the material.

There are some of the following problem in the article that need to be solve.

1.Please introduce the structure of the nanoparticles and the reasons using functionalized COOH ligands.

2.Please introduce the mechanism of iron oxide nanoparticles for photothermal therapy and related literature in introduction section.

3.Please add unit on ''70±14'' of line 88.

4.Please add reference on line 104:’’The high absolute values of the zeta-potential provide strong electrostatic repulsive forces between the nanochains, and result in a good colloidal stability of the suspension in neutral and alkaline conditions.’’

5.Please explain the result on line 258:’’An intriguing result was also observed in case of HeLa-Rab7-GFP cells, where we counted approximately 10 % more colonies in labeled cells.’’

6.Please describe which figure is it at D1 post labeling – Figure 2 on line 271.

7.On line 281 mentioned: Figure 4 E first column, but there are no figure 4E seen in the article.

Author Response

This is a well done research, but some experiments need to be completed.

We sincerely thank the reviewer for the compliment, and also for all suggestions including additional experiments.

Q1. Please show the amount of COOH ligands modified on the material.

A: Since there are no well-known methods for direct quantification of –COOH groups we quantified the primary amines of the RB-nanochains-NH2 by Keiser test (commercial “Keiser test kit” from Sigma Aldrich) following the supplied protocol. The test is based on the reaction of ninhydrin with primary amines, which results in the formation of dark blue color. The spectroscopically determined mean value (in triplicates) of accessible amines on the surface was 145 µmol per gram of the RB-nanochains-NH2 which represent solid estimation of carboxyl moieties on the RB-nanochains-COOH because an excess of SA over the primary amines was used in the synthesis of the RB-nanochains-COOH.

Furthermore, a FTIR-ATR surface characterization analysis was additionally performed in order to confirm the successful reaction of surface amines of the RB-nanochains-NH2 with SA forming free –COOH groups on the surface of the RB-nanochains-COOH. The FTIR-ATR spectra are included as Figure S2 in Supplemental Information.

Finally, we performed an additional zeta potential measurement, although the method is not suitable to confirm the efficiency, of the RB-nanochains-NH2 in order to demonstrate different surface behavior (positive surface charge) at pH 7.4 compared to the one of the RB-nanochains-COOH (negative surface charge) which were prepared from the RB-nanochains-NH2 suggesting the success in binding of SA to the RB-nanochains-NH2. The additional curve of the zeta potential of the RB-nanochains-NH2 was inserted in Figure 1D.

Q2. Please introduce the structure of the nanoparticles and the reasons using functionalized COOH ligands.

A: We thank the reviewer for the comments. We prepared a scheme representing synthesis and functionalization steps and the scheme is inserted in the scheme 1 in, Supplemental Information. There are no direct methods for quantitative assessment of colloidal stability. However, zeta potential, as an indirect method, of the RB-nanochains-COOH shows high absolute values (over ±30 mV) at physiologically relevant pH values reflecting good colloidal stability. In contrary, RB-nanochains are slightly negatively charged (-12 ± 3 mV) while RB-nanochains-NH2 are slightly positively charged (6 ± 5 mV) at pH 7.4 which are both too low absolute values for achieving suitable colloidal stability for biological settings. We therefore applied carboxyl-functionalized nanochains (nanochains-COOH and RB-nanochains-COOH) for our studies.

Q3. Please introduce the mechanism of iron oxide nanoparticles for photothermal therapy and related literature in introduction section.

The introduction was amended accordingly.

Q4.Please add unit on ''70±14'' of line 88.

The unit is the number of nanoparticles in the cluster.

Q5. Please add reference on line 104:’’The high absolute values of the zeta-potential provide strong electrostatic repulsive forces between the nanochains, and result in a good colloidal stability of the suspension in neutral and alkaline conditions.’’

A: We thank the reviewer for the comment. A reference was added: Kralj, S.; Drofenik, M.; Makovec, D. Controlled surface functionalization of silica-coated magnetic nanoparticles with terminal amino and carboxyl groups. J Nanopart Res 2011, 13, 2829-2841.

Q6.Please explain the result on line 258:’’An intriguing result was also observed in case of HeLa-Rab7-GFP cells, where we counted approximately 10 % more colonies in labeled cells.’’

Although the statistical analysis shows that there is a significant difference between nanochains-loaded and unloaded cells, the p-value of this result is only 0.02. As stated in the text, additional tests (namely trypan blue exclusion and the spheroid growth) showed no significant effect of labeling of cancer cells. Taken together these results suggest that there is no intricacy, nor no pro-proliferative effect. Therefore, we rephrased the text.

Q7.Please describe which figure is it at D1 post labeling – Figure 2 on line 271.

All panels in Figure 2 show cells at 1 day post nanoparticle uptake. We have corrected this in the caption. TEM is shown in figure 2F. We corrected this in the text.

Q8. On line 281 mentioned: Figure 4 E first column, but there are no figure 4E seen in the article.

Thank you for pointing out this error. Figure 4A first column.

Reviewer 4 Report

The authors report on an interesting and timely study on the use of nanochains of magnetic nanoparticles coated with silica for photothermal therapy.

I believe the manuscript will be of interest to the readers of Cancers and might be considered for publication after revision of the following points:

In the introduction the authors talk about gold, silver, copper and carbon nanomaterials and mention that these are extremely bio-persistent and can potentially be toxic. The aim of this sentence is to find a better alternative which would be, according to the authors, iron oxide. This sentence needs to be revised. It reads as these materials are of no interest for the biomedical field when in fact some of these nanoparticles are in clinical trials. The references provided are mainly self-citations but do not reflect the real state-of-the-art with these compounds. It is not clear in the text and figure caption whether the TEM image provided in Figure 1 is before or after functionalization. The authors should provide a TEM image of the functionalized material used in the study. The authors talk about colloidal stability. It would be useful to the reader to indicate the stability of these systems, prior and after functionalization, in terms of weight of nanochains per volume. Also indicate for how long are the nanochains stable. Further discussion is needed on why the nanochains suspension leads to an increase of 3.5ºC, whereas after functionalization an increase of 15.7ºC is observed. The physical phenomena that results in such increase is not clear. To how many nanochains does 17pg of iron per cell correspond to? In Figure 5 it seems like the nanochains have lost their structure. High resolution TEM imaging is needed to reveal the structure of the nanochains. This also raises the question on what is the advantage of using nanochains rather than individual nanoparticles of silica coated iron oxide? Is it necessary to use them in the form of nanochains? The authors should include the thermal yield of individual silica particles (with the iron oxide core) before and after functionalization. A reference is needed for the sentence: “Iron oxide nanoclusters, made of Fe3O4 nanocrystals interconnected by amorphous matrix bridges were previously reported as efficient photothermal agents.” The experimental section lacks of details. Even if this work is based on previous studies the volumes and amounts employed should be included. “Slowly added”- over which period of time?; “rigorously stirred” – how?; detail the concentrations employed for zeta-potential measurements and the final mM of KCl after dilution (or indicate the employed volumes to clarify the latter).

Author Response

The authors report on an interesting and timely study on the use of nanochains of magnetic nanoparticles coated with silica for photothermal therapy.

We thank the reviewer for this remark.

I believe the manuscript will be of interest to the readers of Cancers and might be considered for publication after revision of the following points:

In the introduction the authors talk about gold, silver, copper and carbon nanomaterials and mention that these are extremely bio-persistent and can potentially be toxic. The aim of this sentence is to find a better alternative which would be, according to the authors, iron oxide. This sentence needs to be revised. It reads as these materials are of no interest for the biomedical field when in fact some of these nanoparticles are in clinical trials. The references provided are mainly self-citations but do not reflect the real state-of-the-art with these compounds.

We have amended the sentence, which now reads as follows: Photothermal therapy using gold nanoparticles already reached clinical trials (ClinicalTrials.gov Identifiers: NCT01270139 and NCT01436123), yet, as these materials might be extremely bio-persistent [13] and can potentially be toxic [14,15], we suggest alternative bio-compatible materials

It is not clear in the text and figure caption whether the TEM image provided in Figure 1 is before or after functionalization. The authors should provide a TEM image of the functionalized material used in the study.

We are sorry for the unclearness. In the study we only present the biological tests with –COOH functionalized chains, thus the micrograph shows used -COOH functionalized chains. These functional moieties can not be seen on classical TEM, but chemical tests are provided to ascertain successful functionalization

The authors talk about colloidal stability. It would be useful to the reader to indicate the stability of these systems, prior and after functionalization, in terms of weight of nanochains per volume. Also indicate for how long are the nanochains stable.

A: We thank to the reviewer for the comment. There are no very direct methods for quantitative assessment of colloidal stability. However, dynamic light scattering is frequently applied for determination of colloids hydrodynamic size which is, unfortunately, designed for isotropic (spherical) particles. The method is therefore useless for characterization of nanochains due to its highly anisotropic shape. Alternatively, the behavior of colloidal stability in suspension could be estimated by monitoring the spontaneous sedimentation of the colloids over time. The RB-nanochains-COOH remained in suspension (cellular medium) at least three weeks and then only gradual color change could be seen indicating good stability for such a relatively large “colloids”. We added an additional Figure S1 in Supplemental Information. The RB-nanochains-COOH sedimented completely due to gravity in two months but they could be easily redispersed with single gentle shake by hand. Another indirect method is the measurement of the RB-nanochains-COOH zeta potential where high absolute values (over ±30mV) at physiologically relevant pH 7.4 announce good colloidal stability. In contrary, RB-nanochains are slightly negatively charged (-12 ± 3 mV) while RB-nanochains-NH2 are slightly positively charged (6 ± 5 mV) at pH 7.4 which are both too low absolute values for achieving suitable colloidal stability for biological settings. We therefore applied carboxyl-functionalized nanochains (nanochains-COOH and RB-nanochains-COOH) for our studies.

Further discussion is needed on why the nanochains suspension leads to an increase of 3.5ºC, whereas after functionalization an increase of 15.7ºC is observed. The physical phenomena that results in such increase is not clear.

The 3.5°C increase was obtained after magnetic hyperthermia, while a 15.7°C increase was obtained after photo-excitation. We always used the same material, the RB-nanochains-COOH. We have corrected the term nanochains to RB-nanochains-COOH in both cases.

To how many nanochains does 17pg of iron per cell correspond to?

A: The mass of a single nanochain composed of 5 nanoparticle clusters is estimated to be 2.1*10-14 g while iron represents approximately 45% in the composition. Therefore, we could approximate of around 1800 nanochains per cell. However, it is worthy to note that for TEM analysis, cells are sliced in very thin slices (70 nm) slices prepared by ultramicrotome. The slice of the cell is thinner than the length or diameter of the individual nanochain (length estimated to approx. 500 nm and diameter 120 nm) and consequently, if the chains do not lie in the plane parallel to the section, a part of the chain can be cut with the  diamond knife, used to prepare the sample.

In Figure 5 it seems like the nanochains have lost their structure. High resolution TEM imaging is needed to reveal the structure of the nanochains.

As we stated in previous answer, chains in micrographs may appear shorter if they do not lie within the slice. In addition, if the diamond knife, used for ultra-thin-sections preparation, cuts the silica shell in the junction with IONPS, the chains may appear “empty”, but this is just a matter of sample processing. Moreover, as silica provides a very hard shell, sometimes certain clusters might be pulled out of the slice, and a white hole is left, as you can see in SI. Be that as it may, we now provide high magnification TEM images in the Figure 3SI and in New figure 6, to show chains structure on the nano-scale.   We have also added the following text: On the nanoparticles level, the nanochains’ did not undergo any major structural disintegration. Their architecture at day 11 (when the spheroids were fixed and processed for TEM, as evidenced in multicellular spheroids in Figure 5 and 6 and supporting figure S3), remained comparable to nanochains initial structure, which was observed in freshly loaded cells (figure 2B), when the cells were seeded to form spheroids. Indeed over time, the chains are expected to degrade, as silica metabolizes to silicic acid [ref], which is excreted in the urine [ref]. On the other hand, the iron, stemming from iron oxides, integrates the metabolic pathway of endogenous iron [ref].

This also raises the question on what is the advantage of using nanochains rather than individual nanoparticles of silica coated iron oxide? Is it necessary to use them in the form of nanochains? The authors should include the thermal yield of individual silica particles (with the iron oxide core) before and after functionalization.

The advantage of using the chains mainly relates to their magnetic responsiveness, which is much greater compared to individual (unclustered) SPIONs or magnetic clusters. It was our personal choice to use them in the form of chains for the purpose of controlled magnetic manipulation. As their thermal yield stems from iron oxides and not surface modification of silica, -COOH-functionalized chains and unfunctionalized chains have comparable thermal yield.

A reference is needed for the sentence: “Iron oxide nanoclusters, made of Fe3O4 nanocrystals interconnected by amorphous matrix bridges were previously reported as efficient photothermal agents.”

Thank you, for pointing out this editing error. We added the previously omitted reference.

The experimental section lacks of details. Even if this work is based on previous studies the volumes and amounts employed should be included. “Slowly added”- over which period of time?; “rigorously stirred” – how?; detail the concentrations employed for zeta-potential measurements and the final mM of KCl after dilution (or indicate the employed volumes to clarify the latter).

A: The authors thank to the reviewer for her/his comments. We edited the experimental section thoroughly and therefore assured its clarity and precision.

Round 2

Reviewer 1 Report

I read the responses and revised manuscript, and think that most of them are reasonable.

However, still I think that quantitative analysis of cell death is essential for publication to demonstrate the usefulness of nanochains as photothermal agents for therapy. For example, the authors can select one of the assays including MTT, propidium iodide (FACS), or LDH assay.

Other changes are all OK.

Author Response

Reviewer 1:

I read the responses and revised manuscript, and think that most of them are reasonable.

We thank the reviewer for the comment and are glad that he/she is satisfied.

However, still I think that quantitative analysis of cell death is essential for publication to demonstrate the usefulness of nanochains as photothermal agents for therapy. For example, the authors can select one of the assays including MTT, propidium iodide (FACS), or LDH assay.

We fully agree with the reviewer that quantitative analysis of cellular viability is vital for applicability of nanochains for potential photothermal therapy. However, please let us explain our concerns regarding the quantification. As we stated in our manuscript (please refer to lines 349 to 354 of the manuscript after first round revision), tests such as MTT and LDH assays are colorimetric and therefore sensitive to light absorption. It is known that iron oxides absorb light intensively and hence the colorimetric tests are proven to lead to false readouts. Numerous studies reported this fact, and among them, even an article of the Editor of this Special Issue, Prof. Clare Hoskins (Hoskins, C.; Wang, L.; Cheng, W.P.; Cuschieri, A. Dilemmas in the reliable estimation of the in-vitro cell viability in magnetic nanoparticle engineering: which tests and what protocols? Nanoscale research letters 2012), which we cite and warmly recommend to the scientific community.

Propidium iodide test is a good choice and we agree with the reviewer’s suggestion. We used this test in our experiments. The propidium iodide was uptaken by all (100 %) cells after laser exposure, which we state in the text. A representative panel is shown in figure 7 and the video of the process is shown in supporting video. Our laser used in biological experiments was mounted on the multiphoton microscope, where the throughoutput (= number of cells exposed to the system) was relatively low (=limited to a couple of hundreds of cells), because of limited zone that could be illuminated by the miscroscope’s laser.

The reviewer suggests we analyse cell death by FACS analysis of propidium iodide uptake. Unluckily FACS analysis requires a number of at least 10exp5 cells, which can not be obtained with the multiphoton microscope setting. The laser exposure within the multiphoton microscope, which was used to assess the kinetics of cell death by propidium iodide uptake, was limited to the number of cells that could be irradiated during the imaging protocol. While propidium iodide uptake was complete (100% of cells) in all nanoparticle-loaded irradiated cells, this experiment was limited to a couple of hundreds of cells. We are therefore unable to provide flow cytometry analysis, because the latter requires very large populations of cells (typically several hundreds of thousands). Nevertheless, the strength of our method is a direct visualization of the photothermal therapy process, which cannot be obtained by flow cytometer analysis. Moreover, this exposure set-up has other advantages in comparison to FACS, because it allows a direct visualisation of the kinetics of cellular dying (please refer to the supporting video). Finally, to the best of our knowledge, this approach has never been shown in previous studies, and while indeed large cell populations and animal studies are certainly required to ascertain the therapeutic usefulness of our approach, we belive, based on these published empirical results, that our nanoparticles have a great potential.

To underline that we are well aware that other studies now have to be performed, we add the phrase into the text: Further tests, namely tests on large cell populations and animal studies will now be undertaken to go beyond the proof of principle described in this study, and to ascertain the practical therapeutic value of presented nanochains.

Other changes are all OK.

We sincerely appreciate the general positive feedback.

Reviewer 2 Report

The manuscript has addressed most of the comments from the previous review. Here are additional comments.

The manuscript needs to revise the presentation of uptake nanochain quantitation. I understand the estimation of uptaken iron oxide magnetic nanoparticle numbers using the magnetophoresis. However, the results should be particle numbers, NOT pg of Fe. Additionally, the magnetophoresis of nanochain is not the same as nanoparticle because the high anisotropic shape. Therefore, the estimation for the particle number is not likely to be very accurate. Some experiments incubate cells with 5mM Fe. How does the manuscript determine the concentration of Fe in the solution? Additionally, Figure 4B legends state 5mM NP. Is this 5mM Fe, 5mM nanoparticles, or 5mM nanochain? Please carefully go through the manuscript to ensure the consistency.

Author Response

Reviewer 2:

The manuscript needs to revise the presentation of uptake nanochain quantitation. I understand the estimation of uptaken iron oxide magnetic nanoparticle numbers using the magnetophoresis. However, the results should be particle numbers, NOT pg of Fe.

The mass of a single nanochain composed of 5 nanoparticle clusters is estimated to be 2.1*10-14 g. Since iron represents approximately 45% of nanochains, this equals to 0.9*10-14 g of iron per nanochain. As magnetophresis determined an average of 17.3+/-2.6 pg iron per cell, we could approximate of around 1800 +/-270 nanochains per cell. Upon reviewers suggestion, we have added to the text the estimated number of nanochains per cell (highlighted in blue in the revised version round 2).

While we respect the reviewer’s opinion, we do not think that the number of nanoparticles is something that matters more than the mas of iron per cell. Currently, different research groups use different shapes, sizes and structures of nanoparticles, which obviously result in different numbers of nanoparticles per cell. But what can be comparable among different studies, is the iron carried by different nanoparticles.

Please let us give a scholar example to explain why the mass of iron per cell really matters. To make it simple, we could compare different formulations of iron oxides to different formulations of paracetamol. If we say a person ingested 5 pills of paracetamol, is not very informative, as we need to know the concentration/mass of the uptaken active ingredient. It is therefore much more useful to say a person ingested 500 mg of paracetamol, and if we get back to the case of iron oxides, a given mass of iron per cell. Nevertheless, we fully agree with the reviewer on the presentation of our graph in figure 2G, and we have deleted the text in figure 2G, because the extracellular iron concentration as we noted, might have been confusing, as the reviewer pointed out.

Additionally, the magnetophoresis of nanochain is not the same as nanoparticle because the high anisotropic shape. Therefore, the estimation for the particle number is not likely to be very accurate.

We really thank the reviewer for the comment. The issue is completely true and applicable when the nanoparticles are dispersed in suspensions. Conversely, once the nanochains are internalized by cells, they are confined in relatively large endo-lysosomes. These structures are formed at the end of the nanoparticle internalization process. For better illustration, we attach here the micrographs of cells having internalized pristine superparamagnetic nanoparticles, nanoparticles clusters, and nanochains.

At first, nanomaterials (nanoparticles, nanoclusters or nanochains) adhere to cell membrane (please refer to the provided explanation figure in the Microsoft word text; top line). Then, they are internalized in early endosomes, which subsequently fuse together to form late endosomes and endo-lysosomes (Provided figure bottom line). At the end, there are no vesicles with individual nanochains, individual nanoclusters or nanoparticles, but only cells with vesicles, containing large groups of internalized nanostructures showing no shape anisotropy (Provided figure bottom line).

As we are now aware that other readers might have the same reflections as the reviewer, we have added an explanation in the text, which states:

The quantity of iron per cell was determined by magnetophoresis. In the same way as for spherical nanoparticles, once internalized by cells, nanochains end-up in large endo-lysosomal compartments, showing no shape anisotropy. Therefore, in the same way as for other nanoparticles loaded cells, the translational movement of magnetic-nanochains-loaded cells, derives from the overall load, which provides a global magnetic moment of the cell.

Some experiments incubate cells with 5mM Fe. How does the manuscript determine the concentration of Fe in the solution?

The nanochains suspensions dilutions used for cell loading were obtained after diluting a volume of the concentrated stock suspension of RB-nanochains-COOH or Nanochains-COOH (which had a known concentration of 62 mM of iron, as determined by chemical and magnetic measurements) in the RPMI Medium 1640. Cell loading was performed in T-25 flasks containing 2 mL of RPMI and the aliquot containing nanochains suspension, to obtain the final concentration of 5 mM of iron. We added this text to Materials and methods section.

Additionally, Figure 4B legends state 5mM NP. Is this 5mM Fe, 5mM nanoparticles, or 5mM nanochain? Please carefully go through the manuscript to ensure the consistency.

We thank the reviewer for her/his remark. We correct the text throughout accordingly and corrected the figures 2, 3 and 4 in order to improve the clarity and precision.

Round 3

Reviewer 1 Report

Stll, I am not fully satisfied with the manuscript, but the two revisions were prepared well and I will follow the decision of editor and other reviewers.

Reviewer 2 Report

The response has addressed all comments.